# RCPU: Rotation-Constrained Error Compensation for Structured Pruning of Large Language Models

**Shuichiro Haruta, Kazunori Matsumoto, Zhi Li, Yanan Wang, & Mori Kurokawa**
AI Division, KDDI Research, Inc. 2-1-15 Ohara, Fujimino-shi, Saitama 356-8502, Japan
{sh-haruta, mz-matsumoto, zh-li, wa-yanan, mo-kurokawa}@kddi.com

## Abstract

In this paper, we propose a rotation-constrained compensation method to address the errors introduced by structured pruning of large language models (LLMs). LLMs are trained on massive datasets and accumulate rich semantic knowledge in their representation space. In contrast, pruning is typically carried out with only a small amount of calibration data, which makes output mismatches unavoidable. Although direct least-squares fitting can reduce such errors, it tends to overfit to the limited calibration set, destructively modifying pretrained weights. To overcome this difficulty, we update the pruned parameters under a rotation constraint. This constrained update preserves the geometry of output representations (i.e., norms and inner products) and simultaneously re-aligns the pruned subspace with the original outputs. Furthermore, in rotation-constrained compensation, removing components that strongly contribute to the principal directions of the output makes error recovery difficult. Since input dimensions with large variance strongly affect these principal directions, we design a variance-aware importance score that ensures such dimensions are preferentially kept in the pruned model. By combining this scoring rule with rotation-constrained updates, the proposed method effectively compensates errors while retaining the components likely to be more important in a geometry-preserving manner. In the experiments, we apply the proposed method to Llama-7B and Llama-2-13B, and evaluate it on Wiki-Text2 and multiple language understanding benchmarks. The results demonstrate consistently better perplexity and task accuracy compared with existing baselines.

## 1 Introduction

Large language models (LLMs) are driving a rapid wave of transformation and are now being deployed across a wide range of applications, including code assistance, conversational agents, search and summarization, agentic execution, and content generation Jiang et al. (2025); Fan et al. (2024); Zhang et al. (2025); Wang et al. (2024). At the same time, their inference costs in computation and memory remain substantial, creating significant bottlenecks for deployment. In mobile and embedded settings, there is a strong need for model compression techniques that reduce computational cost while preserving task performance Saha & Xu (2025); Girija et al. (2025). Against this backdrop, a variety of efficient methods have been explored, such as quantization, knowledge distillation, and pruning Miao et al. (2025). Among them, *structured pruning* typically removes parameters at the granularity of weight-matrix rows or columns, and in some cases even entire transformer blocks Kim et al. (2024). Such structured removal directly reduces the parameter count and thereby lowers memory usage and inference cost He & Xiao (2024).

Early work targeting LLMs, such as LLM-Pruner, demonstrated the feasibility of compression but tends to rely on downstream fine-tuning to recover high accuracy Ma et al. (2023). Therefore, methods that preserve accuracy without re-training are desirable. WANDA is widely used for unstructured sparsification, and prunes using simple activation-aware heuristics with only a small calibration set without downstream fine-tuning Sun et al. (2024). Its structured variant, Wanda-sp, provides a simple column-pruning importance score and serves as a familiar baseline in structured pruning An et al. (2024). Yet, removing columns inevitably introduces output discrepancies. In

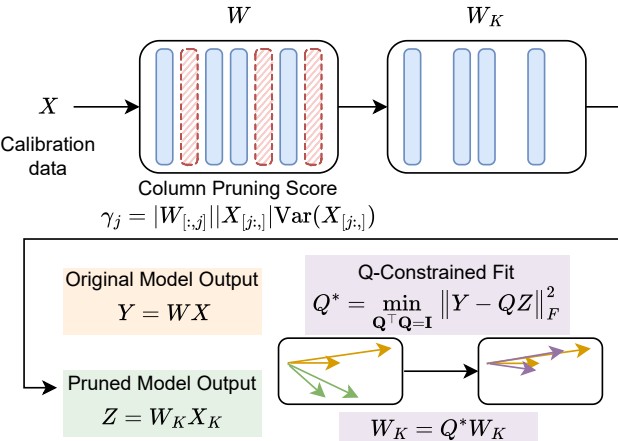

Figure 1: Overview of RCPU framework. Input activations are scored by a variance-aware importance score, less important columns are pruned, and the retained subspace is updated through the rotation-constrained fitting. The pruned output vectors (arrows) are rotated to align with the original output vectors, showing how RCPU compensates for pruning errors.

real deployments, the data available for calibration is often limited. Thus, pruning decisions must be made under sparse observations, and how errors are handled becomes decisive for overall performance. Recent work, FLAP, has shown that compensating the mean component of post-pruning errors with a bias term can be practical and effective An et al. (2024). Nonetheless, input-dependent directional mismatches are not easily addressed by a constant bias. As another possible approach, one might consider least-squares style fitting that directly minimizes output error. However, such a broad parameter update, under limited calibration data, risks overfitting that damages knowledge acquired during pretraining even if using regularization method like Ridge Hoerl & Kennard (1970).

Motivated by these limitations, in this paper, we propose RCPU, a **R**otation-**C**onstrained **P**arameter **U**pdate, to reduce pruning error while preserving the norm and inner-product structure of output representations. Figure 1 shows the overview of RCPU. Compared to general linear least-squares updates, restricting the update to rotations preserves angles and lengths, which helps avoid geometric distortion under small calibration sets. We formulate the alignment between the retained outputs and the original outputs as an Orthogonal Procrustes problem and, for each layer, estimate the optimal rotation and use it to update the parameters. The constraint reduces the update's degrees of freedom, which improves statistical stability and makes it less prone to overfitting. Moreover, since the choice of retained components strongly affects the effectiveness of rotation-constrained compensation, we adopt a simple pruning score that augments weight magnitude and input scale with input variance. By doing this, components contributing to principal output directions are preferentially kept. We combine this scoring rule with rotation-constrained updates, and RCPU effectively compensates errors while retaining the components likely to be more important in a geometry-preserving manner. In the experiments, we apply RCPU to existing LLMs and evaluate it on a variety of language understanding benchmarks. As a result, we demonstrate improvements over existing baselines in both perplexity and task accuracy. The main contributions of this paper are as follows:

- We formulate the compensation method via orthogonal rotation applied immediately after column pruning, and combine it with a simple pruning score that incorporates input variability. We show improvements over existing baselines in post-pruning evaluation.

- The compensation can be inserted directly after Wanda-sp style column pruning, requiring no additional model modifications. It requires no extra architectural changes and adds only modest computation.

## 2  PROBLEM FORMULATION

### 2.1  NOTATION AND SETUP

We consider a linear sub-layer in a transformer block (e.g., attention output projection or multi layer perceptron (MLP) down-projection) with weight matrix $\mathbf{W} \in \mathbb{R}^{d_{\text{out}} \times d_{\text{in}}}$. From a small *calibration* set of $N$ token positions, we record input activations $\mathbf{X} \in \mathbb{R}^{d_{\text{in}} \times N}$ and the corresponding original outputs $\mathbf{Y} = \mathbf{W}\mathbf{X} \in \mathbb{R}^{d_{\text{out}} \times N}$.

We focus on structured pruning methods that drop entire columns of the weight matrix, i.e., column pruning. Structured column pruning selects a binary mask $m \in \{0,1\}^{d_{\text{in}}}$ and keeps the index set $K = \{j \mid m_j = 1\}$ with $|K| = d'$, while $D = \{1, \ldots, d_{\text{in}}\} \setminus K$ denotes the dropped indices. Using $K$ and $D$, we define $\mathbf{W}_K := \mathbf{W}_{[:,K]} \in \mathbb{R}^{d_{\text{out}} \times d'}$, $\mathbf{W}_D := \mathbf{W}_{[:,D]} \in \mathbb{R}^{d_{\text{out}} \times (d_{\text{in}} - d')}$, $\mathbf{X}_K := \mathbf{X}_{[K,:]} \in \mathbb{R}^{d' \times N}$, and $\mathbf{X}_D := \mathbf{X}_{[D,:]} \in \mathbb{R}^{(d_{\text{in}} - d') \times N}$, where $\mathbf{W}_{[:,K]}$ denotes selecting the columns of $\mathbf{W}$ indexed by $K$, and $\mathbf{X}_{[K,:]}$ denotes selecting the rows of $\mathbf{X}$ indexed by $K$. Similarly, $D$ selects the dropped indices. Then the original output decomposes as

$$\mathbf{Y} = \mathbf{W}\mathbf{X} = \underbrace{\mathbf{W}_K \mathbf{X}_K}_{\text{kept}} + \underbrace{\mathbf{W}_D \mathbf{X}_D}_{\text{dropped}}. \tag{1}$$

After pruning, the post-pruning output is

$$\mathbf{Z} = \mathbf{W}_K \mathbf{X}_K \in \mathbb{R}^{d_{\text{out}} \times N}. \tag{2}$$

To compensate the discrepancy without using dropped columns, one possible approach is to update the kept parameters. Concretely, the following regularized optimization problem can be considered:

$$\mathcal{L}(\mathbf{W}^\star) = \|\mathbf{Y} - \mathbf{W}^\star \mathbf{X}_K\|_F^2 + \lambda \|\mathbf{W}^\star - \mathbf{W}_K\|_F^2, \tag{3}$$

where $\lambda$ is regularization hyper-parameter. This corresponds to Ridge regression, in which the updated weights are penalized for deviating from the original ones.

### 2.2  LEAST-SQUARES COMPENSATION

A straightforward way to minimize the error defined in equation 3 is to apply a least-squares fitting. The closed-form solution is

$$\mathbf{W}_K^\star = (\mathbf{Y}\mathbf{X}_K^\top + \lambda \mathbf{W}_K)(\mathbf{X}_K \mathbf{X}_K^\top + \lambda \mathbf{I})^{-1}. \tag{4}$$

**Limitations under limited calibration.** (i) *Geometric distortion:* an unconstrained linear fit may introduce scaling and shear that reduce in-sample error while altering angles and norms in the output space, which can harm generalization beyond the calibration set. Desideratum: preserve the geometry of the outputs as much as possible. (ii) *Limited effectiveness of regularization:* The $\lambda$ values that minimize calibration perplexity do not stabilize the estimator and, in our experiments, often degrade downstream performance. Desideratum: ensure stability in a way that does not depend on regularization tuned purely for calibration set.

These issues motivate restricting the compensation update to geometry-preserving transformations with limited flexibility, while ensuring that the update operates only within the kept subspace.

## 3  RCPU (ROTATION CONSTRAINED PARAMETER UPDATE)

We target the error introduced by structured column pruning in linear sub-layers of a transformer block. After pruning, the kept subspace still carries most of the signal, yet its orientation relative to the original outputs can be misaligned. Our idea is to re-align the orientation by a rotation-constrained parameter update computed from a small calibration set. Restricting the update to rotations preserves norm and inner-product relationships of output representations, helping reduce error while maintaining the pretrained geometry.

This compensation is more effective when the dropped columns do not dominate principal output directions. We therefore combine the rotation with a variance-aware importance score that avoids dropping columns likely to contribute to those directions. Concretely, we extend a common magnitude-and-activation heuristic with an input-variance factor, yielding a simple score that preferentially keeps columns which seem to be relevant for orientation recovery.

## 3.1 ROTATION-BASED COMPENSATION VIA ORTHOGONAL PROCRUSTES

Given $(\mathbf{X}, \mathbf{Y})$ and a pruning mask $K$, we form $\mathbf{Z} = \mathbf{W}_K \mathbf{X}_K$ as in equation 2.

**Optimization problem.** We seek a rotation matrix $\mathbf{Q} \in \mathbb{R}^{d_{\text{out}} \times d_{\text{out}}}$ that aligns the kept output to the original output on calibration data:

$$\mathbf{Q}^\star \;=\; \arg \min_{\mathbf{Q}^\top \mathbf{Q} = \mathbf{I}} \left\| \mathbf{Y} - \mathbf{Q}\mathbf{Z} \right\|_F^2. \tag{5}$$

Equation 5 is known as the classical *Orthogonal Procrustes* problem Golub & Van Loan (2013). It corresponds to the least-squares formulation in equation 3 and equation 4, but with the compensation update restricted to an orthogonal matrix.

**Closed-form solution.** Let $\mathbf{M} = \mathbf{Y}\mathbf{Z}^\top$ and take its singular value decomposition $\mathbf{M} = \mathbf{U}\boldsymbol{\Sigma}\mathbf{V}^\top$, where $\mathbf{U}$ and $\mathbf{V}$ are orthogonal matrices whose columns give the left and right singular vectors and $\boldsymbol{\Sigma}$ is a diagonal matrix containing the singular values of $\mathbf{M}$. Then the minimizer of equation 5 is given by

$$\mathbf{Q}^\star \;=\; \mathbf{U}\mathbf{V}^\top. \tag{6}$$

We update only the kept parameters by rotation as

$$\widetilde{\mathbf{W}}_K \;=\; \mathbf{Q}^\star \mathbf{W}_K. \tag{7}$$

In other words, applying the update makes the new output $\widetilde{\mathbf{W}}_K \mathbf{X}_K = \mathbf{Q}^\star \mathbf{Z}$. Thus, the kept component $\mathbf{Z}$ is explicitly rotated by $\mathbf{Q}^\star$ so that its orientation matches the original output $\mathbf{Y}$. Finally, we replace the sub-layer weight with the compact matrix $\widehat{\mathbf{W}} := \widetilde{\mathbf{W}}_K \in \mathbb{R}^{d_{\text{out}} \times k}$, meaning that columns in $D$ are physically removed. Note that we follow common practice and simply refer to $Q$ as a "rotation" throughout the paper, even though it may occasionally include a reflection.

**Scaled variant.** As a natural extension of the rotation-only solution, we can introduce a single isotropic scaling factor. Although the benefit is expected to be limited, this variant is intuitively reasonable: it preserves the angular structure and norm ratios of the outputs, while also allowing the overall magnitude to be better matched to the original model.

Formally, the optimization problem is defined as

$$(\mathbf{Q}^\star, s^\star) \;=\; \arg \min_{\mathbf{Q}^\top \mathbf{Q} = \mathbf{I},\, s > 0} \left\| \mathbf{Y} - s\,\mathbf{Q}\mathbf{Z} \right\|_F^2. \tag{8}$$

With $\mathbf{M} = \mathbf{Y}\mathbf{Z}^\top = \mathbf{U}\boldsymbol{\Sigma}\mathbf{V}^\top$,

$$\mathbf{Q}^\star = \mathbf{U}\mathbf{V}^\top, \qquad s^\star = \frac{\text{tr}(\boldsymbol{\Sigma})}{\|\mathbf{Z}\|_F^2}, \tag{9}$$

and we set $\widetilde{\mathbf{W}}_K = s^\star \mathbf{Q}^\star \mathbf{W}_K$. This variant rescales all vectors by a common factor $s^\star > 0$. The ordering of vector lengths (within the same set) is invariant.

**Geometric intuition.** Pruning removes the $\mathbf{W}_D \mathbf{X}_D$ term in equation 1, but the kept subspace often still captures much of the useful signal. By restricting the update to rotation (with an optional isotropic scaling), the retained subspace can be re-aligned with the original output geometry while preserving angles and relative norms. This avoids the arbitrary scaling and shear distortions that the least-squares fit may introduce under limited calibration.

## 3.2 VARIANCE-AWARE COLUMN SCORING

To fully exploit rotation-constrained compensation, it is important to retain columns that preserve strong directional information. We therefore assign each input column $j$ with

$$\gamma_j \;=\; \left\| \mathbf{W}_{[:,j]} \right\| \left\| \mathbf{X}_{[j,:]} \right\| \text{Var}(\mathbf{X}_{[j,:]}). \tag{10}$$

The variance term emphasizes columns whose activations fluctuate across calibration tokens, which are more likely to align with dominant output directions. The weight and input norms further bias the

---

**Algorithm 1** Layerwise post-pruning orientation compensation with variance-aware selection

---

**Require:** Calibration tokens; target sub-layers $\mathcal{S}$; pruning ratio $\rho$
1: **for** each transformer layer and each sub-layer $s \in \mathcal{S}$ **do**
2:     **Collect:** record calibration activations $\mathbf{X}$ and original outputs $\mathbf{Y}$
3:     **Score and select columns:**
      • Compute scores $\gamma_j = \|\mathbf{W}_{[:,j]}\| \, \|\mathbf{X}_{[j,:]}\| \, \mathrm{Var}(\mathbf{X}_{[j,:]})$
      • Keep top-$k$ indices $K$ where $k = d_{\mathrm{in}} - \lceil d_{\mathrm{in}}\rho \rceil$
      • Define dropped indices $D = \{1, \ldots, d_{\mathrm{in}}\} \setminus K$
4:     **Form reduced matrices:**
      • $\mathbf{X}_K = \mathbf{X}_{[K,:]}, \mathbf{W}_K = \mathbf{W}_{[:,K]}$, and $\mathbf{Z} = \mathbf{W}_K \mathbf{X}_K$
5:     **Align kept subspace:**
      • Solve equation 5 (or equation 8) for $\mathbf{Q}^\star$ (and $s^\star$)
      • Update kept weights: $\widetilde{\mathbf{W}}_K = \mathbf{Q}^\star \mathbf{W}_K$   (or $s^\star \mathbf{Q}^\star \mathbf{W}_K$)
6:     **Finalize:** replace the weight by $\widehat{\mathbf{W}} = \widetilde{\mathbf{W}}_K$ and remove columns in $D$
7: **end for**

---

score toward columns with inherently larger contributions. This formulation is a natural extension of the WANDA-sp score, which uses only the product of weight and input norms and omits the variance factor. By incorporating variance, our scoring favors columns that not only have large magnitude but also actively contribute under diverse inputs.

Let $\rho \in [0, 1)$ be the pruning ratio for a sub-layer with input width $d_{\mathrm{in}}$; we prune $\lceil d_{\mathrm{in}}\rho \rceil$ columns and keep $k = d_{\mathrm{in}} - \lceil d_{\mathrm{in}}\rho \rceil$ indices with the largest $\gamma_j$.

### 3.3 ALGORITHM AND COMPLEXITY

We apply the procedure layerwise to a designated set of linear sub-layers. Algorithm 1 summarizes the steps. This procedure is *greedy and layerwise*: each layer Procrustes subproblem admits a closed-form minimizer ( equation 6 and equation 9), but the overall routine is not a joint global optimization across the network.

**Complexity.** Per treated sub-layer, computing scores takes $O(d_{\mathrm{in}}(d_{\mathrm{out}} + N))$, forming $\mathbf{Z}$ costs $O(d_{\mathrm{out}}kN)$, and constructing $\mathbf{M} = \mathbf{Y}\mathbf{Z}^\top$ requires $O(d_{\mathrm{out}}^2 N)$. The dominant cost is the SVD of $\mathbf{M} \in \mathbb{R}^{d_{\mathrm{out}} \times d_{\mathrm{out}}}$, which is typically $O(d_{\mathrm{out}}^3)$. Thus the overall complexity is cubic in $d_{\mathrm{out}}$, on par with unstructured pruning methods such as SparseGPT Frantar & Alistarh (2023).

## 4 EXPERIMENTAL RESULTS

### 4.1 SETTINGS

We evaluate RCPU on Llama-7B and Llama-2-13B as the base models Touvron et al. (2023a;b) [1]. As baselines, we use WANDA-sp, a pruning method based on weight magnitude and activation scale; and FLAP, a bias-based error compensation method. In addition, we compare RCPU with SliceGPT. Since SliceGPT does not support Llama-1-7B, we evaluate it on Llama-2-13B with benchmarks.

Following prior work, we use WikiText-2 as the calibration dataset Merity et al. (2016). We perform pruning based on the input channels of o_proj and down_proj, and simultaneously remove the corresponding positions in the other projection matrices. For the attention modules, we prune at the head level. Parameter updates, however, are applied only to o_proj and down_proj. We evaluate pruning ratios of 10%, 20%, and 30%.

Evaluation metrics follow prior studies. We report perplexity (PPL) on WikiText-2, as well as accuracy on a suite of language understanding benchmarks: BoolQ, PIQA, HellaSwag, WinoGrande, ARC-easy, ARC-challenge, and OpenBookQA Clark et al. (2019); Bisk et al. (2020); Zellers et al.

---

[1]Codes are available at `https://github.com/harutaro/rcpu`.

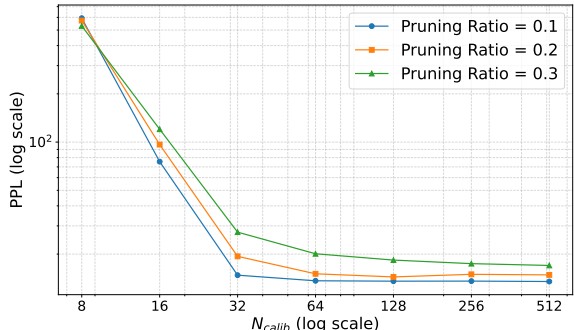

Figure 2: Perplexity versus calibration-set size for RCPU on Llama-7B.

Table 1: Representative PPL ($\downarrow$) of WANDA-sp, FLAP, and RCPU under $N_{\text{calib}} = 128, 512$. The best score in each setting is in **bold**, while the second-best score is underlined. See also Figure 3.

| Method | PR | Llama-7B | | Llama-2-13B | |
|---|---|---|---|---|---|
| | | **128** | **512** | **128** | **512** |
| Original | 0% | 12.4 | 12.4 | 10.98 | 10.98 |
| WANDA-sp | 10% | 14.66 | 14.53 | 12.29 | 12.29 |
| FLAP | 10% | 14.14 | 14.11 | 12.22 | 12.08 |
| RCPU (Rot.) | 10% | 13.55 | 13.48 | **11.65** | **11.57** |
| RCPU (Rot.+Scale) | 10% | **13.52** | **13.45** | 11.66 | **11.57** |
| WANDA-sp | 20% | 16.70 | 16.96 | 14.62 | 14.62 |
| FLAP | 20% | 15.36 | 15.07 | 14.49 | 14.14 |
| RCPU (Rot.) | 20% | **14.40** | 14.83 | 13.12 | 12.75 |
| RCPU(Rot.+Scale) | 20% | 14.55 | **14.81** | **13.07** | **12.72** |
| WANDA-sp | 30% | 24.13 | 26.20 | 61.66 | 63.35 |
| FLAP | 30% | 18.59 | 18.31 | 17.15 | 16.71 |
| RCPU (Rot.) | 30% | 18.35 | 16.99 | 16.99 | 16.01 |
| RCPU (Rot.+Scale) | 30% | **18.21** | **16.91** | **16.88** | **15.96** |

(2019); Sakaguchi et al. (2020); Clark et al. (2018); Mihaylov et al. (2018). These benchmarks cover diverse domains and reasoning types, enabling us to evaluate the model's performance in a comprehensive manner. For evaluation, we adopt the Language Model Evaluation Harness Gao et al. (2024). All experiments were conducted on a single NVIDIA A100 GPU with 80GB memory.

## 4.2 PERPLEXITY

First of all, we conducted experiments regarding PPL, which changes $N_{\text{calib}}$ (The number of calibration samples). Figure 2 reports how the PPL of RCPU varies with the number of calibration samples. We observe that PPL drops rapidly as $N_{\text{calib}}$ increases and becomes roughly stable once $N_{\text{calib}}$ reaches around 64. Based on this trend, we adopt $N_{\text{calib}} = 128$ and 512 as the calibration set sizes for other experiments. While 128 is a common choice in prior work, we additionally include 512, which lies well within the empirically stable region, providing more reliable evaluations.

Table 1 summarizes PPL on WikiText-2 across pruning ratios. As we can see from Table 1, RCPU consistently improves upon WANDA-sp and performs better than FLAP regardless of the number of calibration samples and models. These results show that rotation-constrained updates can be competitive with bias-based compensation. While the advantage is not uniform at all pruning levels, the geometry-preserving nature of rotational transformations helps prevent the distortions that often arise from unconstrained updates. We also compare the scaled variant (Rot.+Scale) to plain rotation. The results show only marginal differences, indicating that introducing a global rescaling does not significantly alter PPL. We believe this is because the rotation already aligns the retained subspace

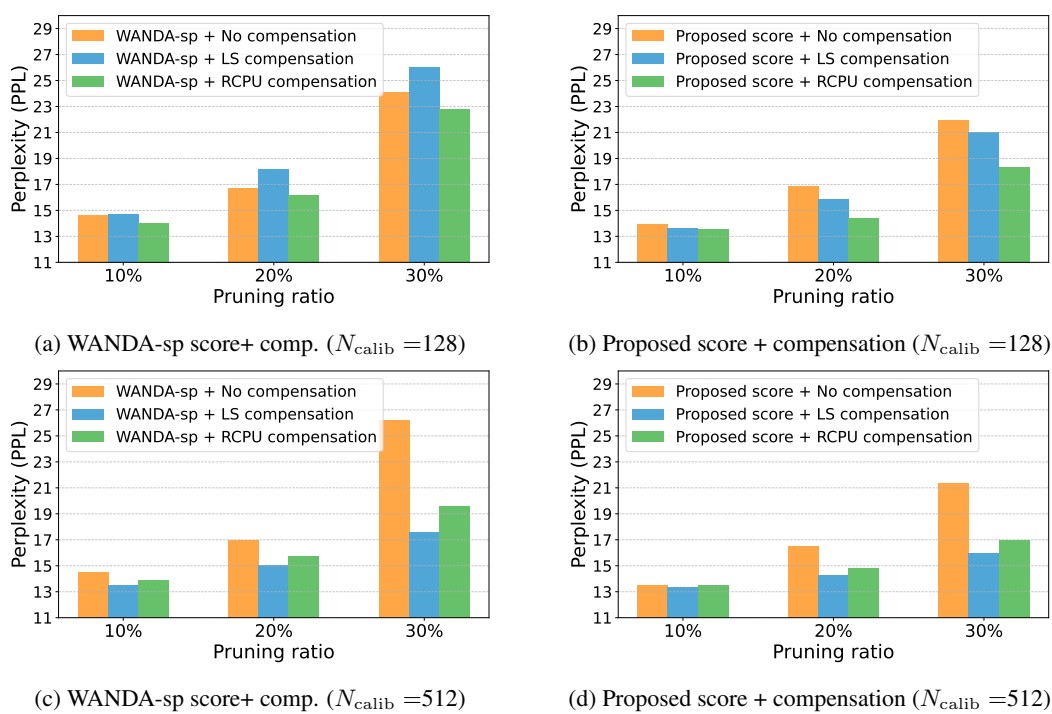

(a) WANDA-sp score+ comp. ($N_{\text{calib}}$ =128)

(b) Proposed score + compensation ($N_{\text{calib}}$ =128)

(c) WANDA-sp score+ comp. ($N_{\text{calib}}$ =512)

(d) Proposed score + compensation ($N_{\text{calib}}$ =512)

Figure 3: PPL vs P.R. ratio for different calibration sizes and compensation methods on Llama-7B.

with the dominant output directions, effectively discarding less informative components. As a result, the overall output norm is well preserved, and an additional scaling factor brings limited benefit.

Next, in Figure 3, we compare the compensation effectiveness of RCPU and LS (Least Square)+Ridge regularization (equation 4). We performed a grid search over $\lambda \in \{10^{-6}, 10^{-5}, \ldots, 10^{6}\}$ and reported the result on best $\lambda$ (See Section 7.1). Figure 3 focuses on the effect of different error-compensation methods (LS or Rot.) in different calibration sizes. From Figure 3a and Figure 3b, we observe that with $N_{\text{calib}} = 128$, the proposed rotation-based compensation achieves the best PPL. In Figure 3a, the regularized LS even worsens PPL, reflecting its tendency to overfit under limited calibration. In contrast, Figure 3c and Figure 3d show that when $N_{\text{calib}} = 512$, both rotation and Ridge-based compensation effectively reduce PPL, but the regularized LS update contributes more strongly to the improvement. Similar trend is observed in Llama-2-13B as shown in Appendix 7.2. Although it is intuitively expected that the effect of the least square fitting becomes larger as the number of calibration samples increases, we emphasize that this does not necessarily translate into better downstream benchmark performance. Indeed, for example in Table 6 in Appendix, LS-based compensations are not the top performer, whereas RCPU often achieves the best accuracy. According to Hastie et al. (2001), the degree of freedom in ridge regression is computed by $d_{\text{out}} \sum_i \frac{\sigma_i^2}{\sigma_i^2 + \lambda}$, where $\sigma$ denotes the singular value of the input matrix. Using this equation and the best $\lambda$, we obtain the degree of freedom values in the range $1.395 \times 10^9$ to $1.578 \times 10^9$ across each pruning ratio. In contrast, the degree of freedom in RCPU is given by $\frac{d_{\text{out}}(d_{\text{out}}-1)}{2}$ since $Q$ is constrained to the orthogonal matrix. Its degree of freedom is $5.36 \times 10^8$, which is smaller than that of LS+Ridge. This implies that, from the standpoint of preserving the pretrained knowledge of LLMs, the rotation-based compensation tends to be more robust. We also highlight that, in the context of LLMs, selecting an appropriate regularization hyper-parameter $\lambda$ can be computationally expensive, as it requires repeatedly computing large matrix inverses for multiple candidate values of $\lambda$. In contrast, our method has no hyper-parameters, avoiding this overhead.

Table 2: Zero-shot accuracy ($\uparrow$) on Benchmark datasets when $N_{\text{calib}} = 128$ on Llama-7B. The best score in each setting is highlighted in **bold**, while the second-best score is underlined.

| Method | P.R. | BoolQ | PIQA | Hella | Wino | ARC-e | ARC-c | OBQA | Mean |
|---|---|---|---|---|---|---|---|---|---|
| Llama-7B (Orig.) | 0% | 75.10 | 78.67 | 76.18 | 70.01 | 72.85 | 44.79 | 44.40 | 66.00 |
| FLAP | 10% | 73.33 | **77.37** | 72.81 | 68.59 | **70.54** | 41.13 | **42.80** | 63.80 |
| WANDA-sp | 10% | **75.17** | 76.82 | **74.43** | 67.01 | 69.53 | **43.43** | 39.60 | 63.71 |
| Prop.Score+LS ($\lambda_{\text{best}}$) | 10% | 72.75 | 75.19 | 71.03 | 68.19 | 64.65 | 39.93 | 39.60 | 61.62 |
| RCPU (Rot.) | 10% | 74.89 | 76.71 | 74.20 | 69.53 | 69.99 | 42.32 | 40.40 | **64.01** |
| RCPU (Rot.+Scale) | 10% | 74.22 | 76.44 | 74.33 | **69.69** | 69.99 | 42.15 | 40.20 | 63.86 |
| FLAP | 20% | 70.45 | 74.91 | 67.42 | 67.64 | **66.79** | 39.00 | **43.00** | 61.32 |
| WANDA-sp | 20% | 69.57 | 75.08 | 69.88 | 65.74 | 65.95 | **40.44** | 38.80 | 60.78 |
| Prop.Score+LS ($\lambda_{\text{best}}$) | 20% | 67.83 | 71.71 | 64.40 | 64.56 | 58.71 | 34.81 | 35.80 | 56.83 |
| RCPU (Rot.) | 20% | 71.50 | 74.81 | **70.32** | **68.43** | 66.58 | 39.93 | 39.40 | **61.57** |
| RCPU (Rot.+Scale) | 20% | **71.77** | 74.97 | 70.25 | 68.03 | 66.71 | 39.93 | 38.80 | 61.49 |
| FLAP | 30% | **66.67** | 71.49 | 59.53 | 61.56 | **59.76** | 34.81 | **39.60** | 56.20 |
| WANDA-sp | 30% | 65.29 | 67.74 | 58.09 | 59.12 | 58.63 | 34.56 | 34.60 | 54.00 |
| Prop.Score+LS ($\lambda_{\text{best}}$) | 30% | 62.54 | 67.08 | 55.55 | 60.77 | 50.13 | 31.06 | 35.00 | 51.73 |
| RCPU (Rot.) | 30% | 61.25 | 70.46 | 62.76 | 62.67 | 58.59 | **34.98** | 37.80 | 55.50 |
| RCPU (Rot.+Scale) | 30% | 65.14 | 70.46 | **62.95** | **64.25** | 58.21 | **34.98** | 38.40 | **56.34** |

Table 3: Zero-shot accuracy ($\uparrow$) on Benchmark datasets when $N_{\text{calib}} = 512$ on Llama-2-13B.

| Method | P.R. | BoolQ | PIQA | Hella | Wino | ARC-e | ARC-c | OBQA | Mean |
|---|---|---|---|---|---|---|---|---|---|
| Llama2-13B (Orig.) | 0% | 80.55 | 79.05 | 79.37 | 72.14 | 77.44 | 49.06 | 45.20 | 68.97 |
| FLAP | 10% | 74.22 | **78.56** | 76.12 | 71.11 | 74.07 | 44.54 | **45.20** | 66.26 |
| SliceGPT | 10% | 62.84 | 77.09 | 71.80 | 71.59 | **76.35** | **49.40** | **45.20** | 64.90 |
| WANDA-sp | 10% | 79.14 | 78.02 | 77.99 | 70.64 | 75.76 | 48.12 | 44.80 | 67.78 |
| Prop.Score+LS ($\lambda_{\text{best}}$) | 10% | 78.78 | 77.69 | 77.56 | 72.30 | 73.99 | 47.53 | 43.40 | 67.32 |
| RCPU (Rot.) | 10% | **79.82** | 78.13 | 78.09 | 72.45 | 75.00 | 47.78 | 44.40 | 67.95 |
| RCPU(Rot.+Scale) | 10% | 79.79 | 78.29 | **78.14** | **72.69** | 75.04 | 47.78 | 44.60 | **68.05** |
| FLAP | 20% | 67.00 | 74.97 | 70.41 | 68.19 | 67.09 | 40.78 | **43.20** | 61.66 |
| SliceGPT | 20% | 52.20 | 71.76 | 63.17 | 67.32 | 70.45 | 43.77 | 41.80 | 58.64 |
| WANDA-sp | 20% | 72.78 | **76.61** | 73.32 | 69.46 | **72.39** | **44.80** | 41.80 | 64.45 |
| Prop.Score+LS ($\lambda_{\text{best}}$) | 20% | 73.12 | 76.28 | 73.02 | 69.93 | 70.50 | 43.86 | 41.20 | 63.99 |
| RCPU (Rot.) | 20% | **73.76** | 76.44 | 73.91 | **70.09** | 71.25 | 44.20 | 42.20 | **64.55** |
| RCPU (Rot.+Scale) | 20% | 73.30 | 76.33 | **73.95** | 69.46 | 71.21 | 43.34 | 41.80 | 64.20 |
| FLAP | 30% | 65.78 | 72.14 | 64.57 | 64.25 | 62.71 | **38.91** | 40.20 | 58.37 |
| SliceGPT | 30% | 38.35 | 66.10 | 52.64 | **66.38** | 56.78 | 35.15 | 40.00 | 50.77 |
| WANDA-sp | 30% | 61.99 | 62.68 | 36.09 | 51.07 | 41.54 | 25.60 | 28.80 | 43.97 |
| Prop.Score+LS ($\lambda_{\text{best}}$) | 30% | **66.61** | 72.96 | 64.62 | 65.04 | 66.62 | 36.77 | 40.40 | 59.00 |
| RCPU (Rot.) | 30% | 64.37 | **73.72** | 66.22 | 64.88 | 67.05 | 38.31 | **42.80** | 59.62 |
| RCPU (Rot.+Scale) | 30% | 65.17 | 73.67 | **66.69** | 64.88 | **67.30** | 38.23 | 42.60 | **59.79** |

## 4.3 BENCHMARK

Table 2 and Table 3 reports accuracy on seven language understanding benchmarks on Llama-7B and Llama-2-13B. Overall, performance degrades as the pruning ratio increases. RCPU achieves higher mean accuracy than FLAP across all pruning levels, indicating that geometry-preserving compensation can be more effective than bias-only correction. Comparing Table 2 with Table 6, the scaled variant performed better and ranked as the best baseline more often in the 512-sample setting than in the 128-sample setting. Intuitively, the additional samples stabilize the norm statistics of the unpruned versus pruned outputs, allowing the global scale $s^\star$ to more effectively restore the original

Table 4: PPL under different pruning ratios and compensation targets (Llama-7B, $N_{\text{calib}} = 128$).

| Pruning ratio | No compensation | o_proj only | down_proj only | Both updated |
|---|---|---|---|---|
| 10% | 13.96 | 13.61 | 13.62 | 13.55 |
| 20% | 16.85 | 15.47 | 15.76 | 14.40 |
| 30% | 21.94 | 18.91 | 20.22 | 18.35 |

Table 5: Size of the pruned model and time required for pruning (Llama-7B).

| Pruning ratio | # of parameters | Memory size | Time required for pruning a layer |
|---|---|---|---|
| 0% (FP32) | 6.73B | 25,705MiB | - |
| 10% | 6.10B | 23,295MiB | 8.36s |
| 20% | 5.47B | 20,875MiB | 8.90s |
| 30% | 4.84B | 18,456MiB | 9.20s |

magnitude. Regarding Table 3, SliceGPT transforms the entire model into an equivalent structure using an orthogonal matrices, and then performs row or column deletion in a single global step. Due to this property of applying a global transformation followed by global pruning, we expect that, at high pruning ratios, a mismatch arises between the input distributions assumed by each layer and the actual input distributions after pruning. This mismatch is also expected to accumulate across layers and thus we believe accuracy tends to degrade at high pruning ratios. In contrast, RCPU optimizes the pruning-induced error layer by layer. Thus, even at high pruning ratios, each layer is more likely to maintain representations close to the inputs it assumes, which we believe leads to better benchmark performance. Looking at individual tasks, HellaSwag and WinoGrande show relatively stronger performance with RCPU. These tasks require contextual consistency and pronoun resolution, both of which are sensitive to orientation shifts in the representation space. The benefit observed here aligns with the perplexity improvements reported earlier, suggesting that rotation-constrained updates help preserve the structural properties of output representations that underlie these tasks.

## 4.4 ANALYSIS

**Where to apply rotation?**  In order to clarify which parts are effective to apply RCPU, we conducted ablation study that changes the module to apply the proposed compensation. Table 4 reports PPL when rotation-based compensation is applied to different projection sub-layers. First, applying compensation to either o_proj or down_proj alone improves PPL compared to the uncompensated baseline. This indicates that the kept subspace indeed contains recoverable signal, and aligning it to the original outputs partially restores the lost information. Second, updating o_proj is consistently more effective than updating down_proj. A plausible explanation lies in the forward order of computation in transformer blocks: attention is followed by the MLP. Misalignment at o_proj propagates directly into the subsequent MLP input, thereby amplifying its negative effect. Correcting the orientation earlier at o_proj provides the MLP with already aligned features, reducing the burden of later layers. In contrast, compensating only down_proj cannot undo the upstream misalignment originating from o_proj, and thus achieves a smaller gain. Finally, applying compensation to both o_proj and down_proj yields the largest improvement, suggesting that errors at the two sites are complementary. Moreover, the benefit becomes larger at higher pruning ratios, where the retained subspace is smaller and orientation recovery plays a more critical role.

**Efficiency**  Table 5 summarizes the number of parameters, memory usage, and pruning time per layer at different pruning ratios. As expected, both the parameter count and memory decrease monotonically as the pruning ratio increases, confirming the resource savings of structured pruning. In contrast, pruning time exhibits a counter-intuitive trend. While the dominant computation is the SVD of $M = Y Z^\top \in \mathbb{R}^{d_{\text{out}} \times d_{\text{out}}}$, which incurs a constant $O(d_{\text{out}}^3)$ cost, the computation of $Z = W_K X_K$ depends on the number of kept columns $k$ (Section 3.3). In principle, a larger pruning ratio (smaller $k$) should make this step cheaper. However, in practice, we observed that pruning time becomes

slightly longer at higher ratios. This is because when $k$ is small, $W_K X_K$ results in a tall and narrow matrix multiplication, which GPU libraries (e.g., cuBLAS) handle less efficiently than more square-shaped matrices. Similar behavior has been reported in prior work Rivera et al. (2021). Notably, pruning completes within 10 seconds per layer, suggesting that the proposed method remains practically applicable even to larger-scale models.

## 5 RELATED WORK

**Model compression in LLMs** Large language models (LLMs) incur substantial computational and memory costs, motivating the development of compression techniques. Among the most widely studied approaches are *quantization*, which lowers precision to improve efficiency Lang et al. (2024), and *distillation*, which transfers knowledge from a large teacher to a smaller student model Yang et al. (2024). In this work, we focus on *pruning*, which removes unnecessary parameters.

**Unstructured pruning** Unstructured pruning accelerates inference by sparsifying weight matrices. SparseGPT Frantar & Alistarh (2023) enables one-shot pruning of LLMs via an efficient second-order update. WANDA Sun et al. (2024) introduces an activation-aware importance score that works with only a small calibration set. There also exist approaches that pursue higher accuracy, such as AlphaPruning Lu et al. (2024), which varies the pruning ratio across layers. While effective, unstructured methods do not actually reduce parameter count. Their memory and speed benefits depend on specialized hardware supports, and thus they are generally unsuitable for small devices.

**Structured Pruning** Structured pruning removes parameters at the level of rows, columns, or blocks, directly shrinking model size and memory footprint. LLM-Pruner Ma et al. (2023) demonstrates structured pruning for LLMs but typically requires downstream fine-tuning, and Shortened-LLaMA Kim et al. (2024) prunes depth (layers) with retraining. In contrast, methods such as Wanda-sp and FLAP are applicable without additional retraining An et al. (2024): Wanda-sp extends WANDA's activation-aware rule to column pruning, while FLAP compensates post-pruning errors via a bias term. SliceGPT Ashkboos et al. (2024) leverages a computational invariance of RMSNorm-connected transformers: by applying an orthogonal reparameterization (derived via principal component analysis), the model is reformulated into a rotated basis where entire rows and columns can be deleted while preserving functional equivalence. Rotation has also been used to improve prunability in other forms: RotPruner Chen & Wang (2025) learns layer-wise orthogonal transforms to obtain pruning-friendly parameterizations, and DenoiseRotator Gu et al. (2025) trains rotations that concentrate importance scores before pruning. These methods use rotation as an additional parameterization to facilitate pruning before or during parameter removal. Unlike these rotation-learning or slicing-based approaches, RCPU focuses on reducing the pruning error after structured column removal through an analytically derived orthogonal compensation. Our work is closer to FLAP, trying to minimize pruning-induced output error and achieve better performance.

## 6 CONCLUSION

In this paper, we proposed RCPU, a rotation-constrained error compensation method, for structured pruning of large language models. By formulating post-pruning recovery as an Orthogonal Procrustes problem, our approach aimed to preserve the geometry of output representations while re-aligning the retained subspace to the original outputs. To complement this update, we adopted a variance-aware importance score that preferentially retains columns contributing to principal output directions, thereby enhancing the effectiveness of rotation-constrained compensation. Through experiments on Llama-7B and Llama-2-13B, we demonstrated that RCPU improves perplexity and task accuracy across pruning ratios, outperforming existing baselines. The improvements were particularly pronounced at higher pruning levels, indicating that geometry-preserving updates become increasingly critical as the retained subspace shrinks. Moreover, we showed the method requires no additional architectural changes and only modest computational overhead, making it practically applicable to large-scale deployments.

Overall, our findings suggest that incorporating geometric constraints into error compensation for pruning can be a promising direction. We hope that this framework stimulates further investigation into pruning-aware model updates that balance statistical stability and computational efficiency.

## ETHICS AND REPRODUCIBILITY STATEMENT

**Disclosure of AI assistance.** We used a large language model to edit and polish the manuscript text. All research ideas, methods, and experiments were conducted solely by the authors.

**Data usage and privacy.** All calibration and evaluation datasets used in this work are publicly available and contain no sensitive personal information. We used the data under their respective licenses, made no attempts at re-identification, and did not store or share model inputs and outputs beyond the scope of calibration and evaluation.

**Environmental impact.** This study adds only a small incremental compute footprint: calibration consists of forward passes plus one small SVD per targeted sub-layer. We did not perform gradient-based fine-tuning in our experiments. At deployment time, structured pruning reduces parameter count and effective FLOPs, which can lower inference cost under comparable hardware and batching conditions.

**Fairness and safety.** Structured pruning can alter performance unevenly across tasks, domains, or languages. We therefore evaluate on diverse benchmarks and report per-task metrics to surface potential regressions. No safety-critical deployment is claimed.

**Reproducibility.** All experimental settings and tools are described in the main text, including the models and datasets and the calibration setup.

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

## 7 APPENDIX

### 7.1 HOW TO DETERMINE THE BEST $\lambda$

We select the regularization strength $\lambda_{\text{best}}$ for the Ridge+LS baseline as the value that minimizes the perplexity on the same calibration data used for computing the compensation. This choice is motivated by the following reasons. First, the calibration set is very small, consisting of only 128–512 samples. Partitioning this already limited set into separate train and validation subsets would render the estimation of $\lambda$ statistically unstable. Indeed, in our experiments on Llama-2-13B, even with 512 calibration samples, the perplexity varies substantially across different $\lambda$ values (see Section 7.4). Introducing an additional split would further reduce the effective sample size and increase estimation variance. Second, RCPU itself also relies solely on the calibration samples and does not use a separate validation set. Using validation only for the Ridge baseline would therefore introduce an inconsistency in the comparison protocol.

### 7.2 PPL COMPARISON IN LLAMA-2-13B

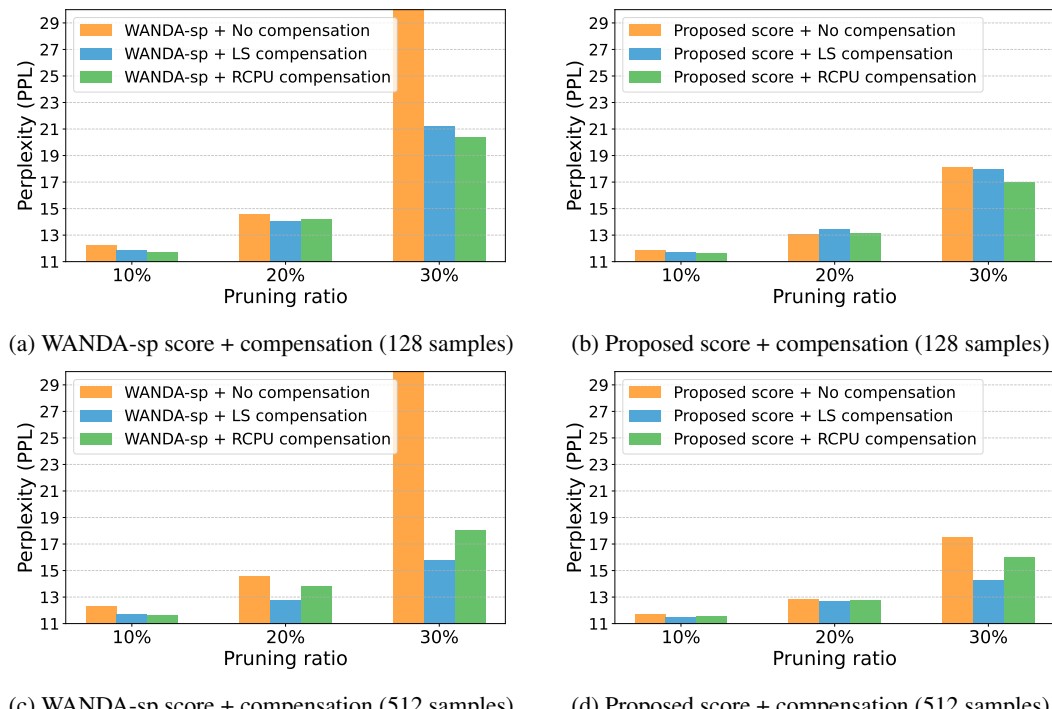

(a) WANDA-sp score + compensation (128 samples)

(b) Proposed score + compensation (128 samples)

(c) WANDA-sp score + compensation (512 samples)

(d) Proposed score + compensation (512 samples)

Figure 4: PPL vs P.R. for different calibration sizes and compensation methods on Llama-2-13B.

### 7.3 OTHER BENCHMARKS AND MODELS

Table 6: Llama-1-7B calib 512

| Method | P.R. | BoolQ | PIQA | Hella | Wino | ARC-e | ARC-c | OBQA | Mean |
|---|---|---|---|---|---|---|---|---|---|
| Llama-7B (Original) | 0% | 75.10 | 78.67 | 76.18 | 70.01 | 72.85 | 44.79 | 44.40 | 66.00 |
| FLAP | 10% | 74.46 | **77.75** | 73.05 | 68.19 | 70.66 | 41.98 | **43.20** | 64.18 |
| WANDA-sp | 10% | 75.35 | 76.82 | 74.14 | 68.51 | **71.17** | **44.20** | 38.80 | 64.14 |
| Prop.Score+LS ($\lambda_{\text{best}}$) | 10% | 75.02 | 76.77 | 73.33 | **68.59** | 70.75 | 42.49 | 39.00 | 63.71 |
| RCPU (Rot.) | 10% | 76.06 | 76.88 | 74.45 | 68.27 | 70.96 | 42.92 | 41.40 | 64.42 |
| RCPU(Rot.+Scale) | 10% | **76.18** | 76.77 | **74.50** | 68.43 | 70.83 | 43.00 | 41.80 | **64.50** |
| FLAP | 20% | 71.07 | 75.24 | 68.46 | 66.93 | **68.06** | 40.27 | **41.00** | 61.58 |
| WANDA-sp | 20% | 66.94 | 75.14 | 69.85 | 66.06 | 64.98 | 40.53 | 38.40 | 60.27 |
| Prop.Score+LS ($\lambda_{\text{best}}$) | 20% | 71.71 | 74.76 | 69.70 | 67.48 | 66.29 | 37.97 | 39.20 | 61.02 |
| RCPU (Rot.) | 20% | 72.87 | **76.12** | 70.93 | **67.96** | 68.01 | 39.93 | 39.40 | 62.17 |
| RCPU (Rot.+Scale) | 20% | **73.43** | 75.52 | 70.80 | 67.80 | **68.06** | **40.61** | 39.60 | **62.26** |
| FLAP | 30% | **67.00** | 71.38 | 60.85 | 63.22 | 58.88 | 34.39 | **40.20** | 56.56 |
| WANDA-sp | 30% | 58.44 | 68.28 | 56.69 | 56.35 | 56.14 | 33.70 | 36.00 | 52.23 |
| Prop.Score+LS ($\lambda_{\text{best}}$) | 30% | 64.53 | 71.11 | 62.32 | 61.96 | 59.76 | 33.79 | 37.20 | 55.81 |
| RCPU (Rot.) | 30% | 65.47 | 71.65 | **64.47** | 62.19 | 61.95 | 36.18 | 36.60 | 56.93 |
| RCPU (Rot.+Scale) | 30% | 65.29 | **71.71** | 64.41 | **63.30** | **63.13** | **36.35** | 37.60 | **57.40** |

Table 7: Llama-2-13B calib 128

| Method | P.R. | BoolQ | PIQA | Hella | Wino | ARC-e | ARC-c | OBQA | Mean |
|---|---|---|---|---|---|---|---|---|---|
| Llama2-13B (Original) | 0% | 80.55 | 79.05 | 79.37 | 72.14 | 77.44 | 49.06 | 45.20 | 68.97 |
| FLAP | 10% | 70.98 | **78.13** | 75.72 | 69.53 | 72.85 | 45.22 | **44.80** | 65.32 |
| SliceGPT | 10% | 68.10 | 76.00 | 71.17 | 71.59 | 75.25 | 48.29 | 43.20 | 64.80 |
| WANDA-sp | 10% | 78.01 | 77.86 | **77.98** | 71.03 | **75.88** | **48.81** | 44.60 | **67.74** |
| Prop.Score+LS ($\lambda_{\text{best}}$) | 10% | 78.47 | 77.26 | 77.34 | 71.27 | 73.02 | 46.42 | 43.00 | 66.68 |
| RCPU(Rot.) | 10% | **78.96** | 77.75 | 77.78 | 71.82 | 74.11 | 46.24 | 43.60 | 67.18 |
| RCPU(Rot.+Scale) | 10% | 78.89 | 78.12 | 77.75 | **71.90** | 73.73 | 46.16 | 44.00 | 67.22 |
| FLAP | 20% | 69.48 | 74.65 | 69.59 | 67.17 | 66.67 | 40.87 | 42.60 | 61.58 |
| SliceGPT | 20% | 44.89 | 71.55 | 62.78 | 68.35 | 67.42 | 42.06 | 41.40 | 56.92 |
| WANDA-sp | 20% | **73.21** | **76.99** | **73.06** | 69.14 | **71.17** | **44.54** | 42.80 | **64.42** |
| Prop.Score+LS ($\lambda_{\text{best}}$) | 20% | 71.77 | 74.70 | 72.30 | 69.53 | 69.28 | 42.41 | 41.40 | 63.06 |
| RCPU(Rot.) | 20% | 72.93 | 76.06 | 73.03 | **70.32** | 70.03 | 43.08 | **43.20** | 64.09 |
| RCPU(Rot.+Scale) | 20% | 72.50 | 75.95 | 72.93 | 70.24 | 69.52 | 43.08 | 42.00 | 63.74 |
| FLAP | 30% | 64.07 | 71.00 | 63.33 | 63.54 | **62.92** | **39.68** | **40.80** | 57.91 |
| SliceGPT | 30% | 39.08 | 65.13 | 52.29 | 65.43 | 53.45 | 36.69 | 39.20 | 50.18 |
| WANDA-sp | 30% | 62.01 | 63.49 | 35.69 | 49.64 | 42.21 | 25.76 | 28.40 | 43.89 |
| Prop.Score+LS ($\lambda_{\text{best}}$) | 30% | 61.19 | 69.21 | 60.20 | 61.17 | 57.87 | 33.19 | 37.00 | 54.26 |
| RCPU(Rot.) | 30% | **66.48** | **72.85** | **65.37** | **65.66** | 61.53 | 36.68 | 40.40 | **58.42** |
| RCPU(Rot.+Scale) | 30% | 66.45 | 72.79 | 64.93 | 64.95 | 61.57 | 36.26 | 39.80 | 58.11 |

Table 8: PPL comparison in Vicuna-7B under $N_{\text{calib}} = 128, 512$.

| Method | PR | Vicuna-7B | |
|---|---|---|---|
| | | **128** | **512** |
| Original | 0% | 16.24 | 16.24 |
| Prop.Score+LS ($\lambda_{\text{best}}$) | 10% | 19.10 | 17.58 |
| RCPU (Rot.) | 10% | 17.54 | 17.26 |
| RCPU (Rot.+Scale) | 10% | **17.52** | **17.22** |
| Prop.Score+LS ($\lambda_{\text{best}}$) | 20% | 20.53 | **18.54** |
| RCPU (Rot.) | 20% | 19.62 | 18.99 |
| RCPU (Rot.+Scale) | 20% | **19.59** | 18.93 |
| Prop.Score+LS ($\lambda_{\text{best}}$) | 30% | 26.54 | **20.52** |
| RCPU (Rot.) | 30% | 23.11 | 21.55 |
| RCPU (Rot.+Scale) | 30% | **22.91** | 21.52 |

Table 9: Vicuna-7B calib 128

| Method | P.R. | BoolQ | PIQA | Hella | Wino | ARC-e | ARC-c | OBQA | Mean |
|---|---|---|---|---|---|---|---|---|---|
| Vicuna-7B (Original) | 0% | 80.92 | 77.31 | 73.76 | 69.38 | 71.25 | 45.90 | 45.00 | 66.22 |
| Prop.Score+LS ($\lambda_{\text{best}}$) | 10% | 71.22 | 72.63 | 68.55 | 66.54 | 65.49 | 39.42 | 38.60 | 60.35 |
| RCPU (Rot.) | 10% | **78.26** | **76.33** | **72.39** | **68.67** | 71.25 | 44.62 | 42.80 | **64.90** |
| RCPU(Rot.+Scale) | 10% | 78.04 | 76.22 | 72.29 | 68.51 | **71.38** | **44.80** | **43.00** | 64.89 |
| Prop.Score+LS ($\lambda_{\text{best}}$) | 20% | 66.67 | 73.18 | 65.56 | 63.93 | 64.90 | 39.25 | 36.60 | 58.58 |
| RCPU (Rot.) | 20% | **70.95** | 73.56 | **68.37** | 66.14 | 66.20 | **41.21** | 40.80 | 61.03 |
| RCPU (Rot.+Scale) | 20% | 70.21 | **73.88** | 68.35 | **66.77** | **66.33** | 40.87 | **41.40** | **61.12** |
| Prop.Score+LS ($\lambda_{\text{best}}$) | 30% | 52.23 | 65.07 | 53.62 | 59.12 | 54.97 | 31.74 | 36.60 | 50.48 |
| RCPU (Rot.) | 30% | 62.66 | **69.91** | 60.34 | 62.59 | 61.28 | 35.84 | **40.00** | 56.09 |
| RCPU (Rot.+Scale) | 30% | **63.70** | 69.70 | **60.79** | **62.67** | **61.87** | **37.20** | 40.60 | **56.65** |

Table 10: Vicuna-7B calib 512

| Method | P.R. | BoolQ | PIQA | Hella | Wino | ARC-e | ARC-c | OBQA | Mean |
|---|---|---|---|---|---|---|---|---|---|
| Vicuna-7B (Original) | 0% | 80.92 | 77.31 | 73.76 | 69.38 | 71.25 | 45.90 | 45.00 | 66.22 |
| Prop.Score+LS ($\lambda_{\text{best}}$) | 10% | 78.10 | **76.66** | **71.62** | 67.88 | 70.92 | 43.94 | 41.60 | 64.39 |
| RCPU (Rot.) | 10% | 78.35 | 76.44 | 72.60 | **68.43** | 71.93 | 44.80 | **43.00** | 65.08 |
| RCPU(Rot.+Scale) | 10% | **78.44** | 76.61 | 72.54 | 67.88 | **72.01** | **45.48** | 42.80 | **65.11** |
| Prop.Score+LS ($\lambda_{\text{best}}$) | 20% | **71.90** | 74.32 | 68.19 | 64.01 | 67.76 | 41.89 | 40.40 | 61.21 |
| RCPU (Rot.) | 20% | 70.55 | **74.86** | 69.01 | 65.90 | **67.93** | 41.89 | **41.40** | **61.65** |
| RCPU (Rot.+Scale) | 20% | 71.22 | 74.21 | **69.35** | **65.98** | 67.42 | 41.55 | 40.80 | 61.50 |
| Prop.Score+LS ($\lambda_{\text{best}}$) | 30% | **65.75** | **70.89** | 61.43 | 63.06 | 62.33 | 37.80 | **40.00** | 57.32 |
| RCPU (Rot.) | 30% | 64.89 | 70.67 | **62.55** | 63.61 | 63.01 | **38.40** | 39.80 | **57.56** |
| RCPU (Rot.+Scale) | 30% | 65.29 | 70.67 | 62.38 | **63.69** | **63.34** | 37.63 | 39.20 | 57.46 |

## 7.4 PPL vs $\lambda$ in Least Square with Ridge

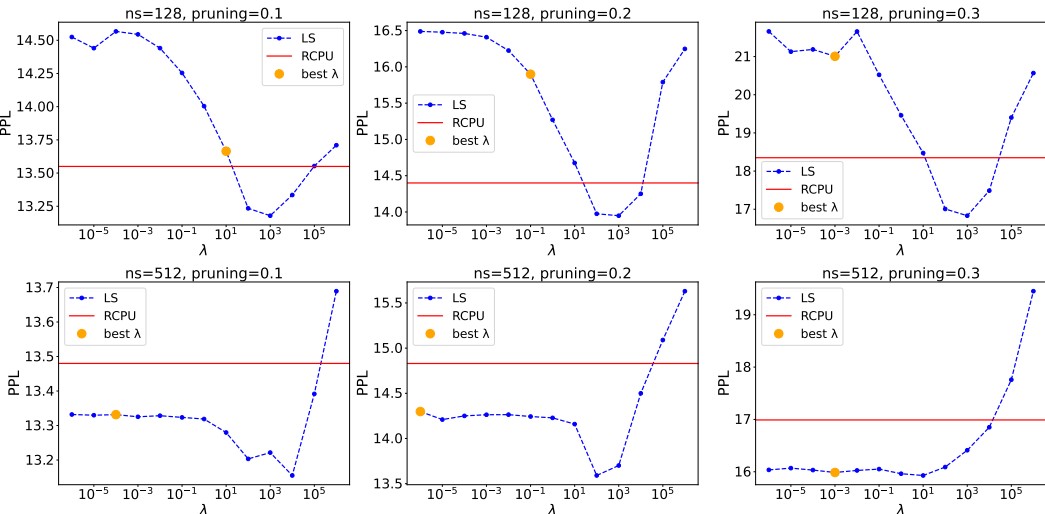

Figure 5: Llama-7B: PPL vs $\lambda$ in LS+Ridge (Pruned by proposed score)

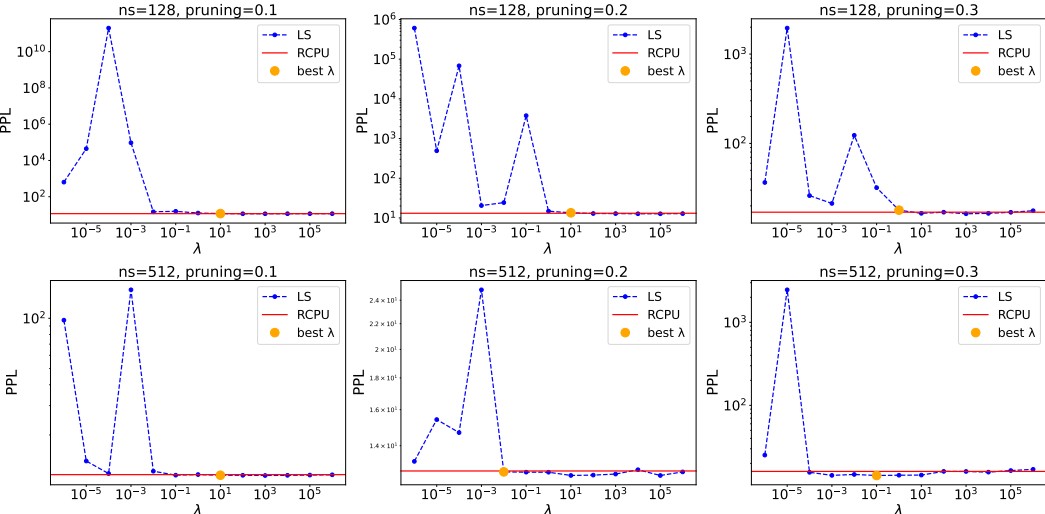

Figure 6: Llama-2-13B: PPL vs $\lambda$ in LS+Ridge (Pruned by proposed score)

Figure 5 and Figure 6 show the PPL versus $\lambda$ in Llama-7B and Llama-2-13B models. Overall, the best $\lambda$ varies widely depending on the model type, pruning ratio, and the number of calibration samples. This suggests that finding an optimal $\lambda$ in a reliable manner is inherently difficult. For Llama-7B, the best $\lambda$ chosen on the calibration set does not yield the best test-set performance, indicating insufficient generalization. For Llama-2-13B, the PPL becomes unstable for certain $\lambda$ values (especially near zero). This instability is particularly pronounced when the calibration size is 128, which likely reflects the severe mismatch between the number of parameters and the amount of available calibration data. The best $\lambda$ achieves performance comparable to RCPU when using 128 samples, and slightly better than RCPU when using 512 samples.

However, across models and calibration sizes, RCPU consistently outperforms LS+Ridge on downstream tasks (Table 2, Table 3, Table 6, Table 7). This indicates that LS+Ridge compensation achieves some degree of in-domain generalization but fails to generalize out-of-domain. From the perspective of preserving the pretrained knowledge of the LLM, RCPU provides a more robust form of compensation.

