# OpenReview forum: "RCPU: Rotation-Constrained Error Compensation for Structured Pruning of Large Language Models"
_ICLR.cc/2026/Conference — ICLR 2026 Poster_

### Official Review · Reviewer_f6Ag · 2025-10-22

**Soundness:** 3
**Presentation:** 3
**Contribution:** 3
**Rating:** 6
**Confidence:** 4

**Summary:**

In this paper, the authors aim to mitigate pruning errors in structured pruning for LLMs. They propose a rotation-constrained compensation scheme to preserve geometric information and minimize mismatches during parameter updates. Furthermore, a variance-aware importance score is introduced to maintain the principal output directions. Experiments conducted on LLaMA-7B demonstrate the effectiveness of the proposed method.

Structured pruning is valuable for the practical deployment of LLMs. I lean toward a marginal accept for this paper. If the authors address my concerns during the rebuttal phase, I would like to raise my score.

**Strengths:**

Originality:
The paper introduces a rotation-based structured pruning method for LLMs. While the concept of rotation-based pruning is not new, this work offers an alternative approach to structured pruning with clear practical value for real-world deployment.

Quality:
The paper is well-written and logically organized. The motivation is clearly articulated, and the problem formulation is well-grounded. Both the proposed algorithm and its accompanying analysis are reasonable. The experimental results support the authors’ claims.

Clarity:
The paper is clearly structured and easy to follow. The presentation of content and formulas are well-explained.

Significance:
This work focuses on recovering pruning errors in structured pruning for LLMs, which is a meaningful contribution that enhances the practical deployability of large-scale models.

**Weaknesses:**

Main Comments:
1. The proposed rotation idea is very similar to prior works such as RotPruner[1] and DenoiseRotator[2]. Although this paper applies the rotation concept to structured pruning, the aforementioned methods were designed for unstructured and semi-structured pruning. The authors should discuss these related studies in the Introduction and Related Work sections to clarify their distinctions.

2. The experiments are limited to LLaMA-7B. It would strengthen the paper to include results on larger models (e.g., 13B–70B) to verify scalability and generality.

3. It would be helpful to discuss whether the proposed rotation scheme can also be extended to unstructured or semi-structured pruning scenarios.

Minor Comments:
1. Lines 166–167: “an rotation-constrained parameter update” should be corrected to “a rotation-constrained parameter update.”


[1] RotPruner: Large Language Model Pruning in Rotated Space.
[2] DenoiseRotator: Enhance Pruning Robustness for LLMs via Importance Concentration. NeurIPS 2025

**Questions:**

1. The proposed rotation-based pruning method appears conceptually similar to RotPruner and DenoiseRotator. Could the authors clarify the key differences and novel aspects of their approach compared to these prior works? In particular, how does the proposed rotation constraint contribute uniquely to structured pruning beyond what has been achieved in unstructured or semi-structured settings?

2. The experiments are conducted only on LLaMA-7B. Do the authors expect the proposed method to scale effectively to larger models such as LLaMA-13B or LLaMA-70B?

3. Can the proposed rotation scheme be generalized to unstructured or semi-structured pruning?

4. Please correct the minor grammatical issue in Lines 166–167 (“an rotation-constrained” → “a rotation-constrained”).

---

> ### Author Response · Authors · 2025-11-26
> **[Weakness2, Question2] Scalability to the larger model**
>
> Thank you for your constructive comments. As the reviewer f6Ag pointed out, we can highlight the strength of RCPU by adding the evaluation results that RCPU can scale to larger models.
>
> To evaluate scalability beyond Llama-7B, we additionally tested RCPU on Llama-2-13B using PPL and benchmark tasks.
> As shown in  [Table1](https://anonymous.4open.science/r/anonymous-llm-pruning-D884/table1.png), [Table3](https://anonymous.4open.science/r/anonymous-llm-pruning-D884/table3.png) and [Table7](https://anonymous.4open.science/r/anonymous-llm-pruning-D884/table7.png), RCPU maintains superior performance and follows the same trend as on Llama-7B, while absolute scores increase due to the stronger capability of the larger model.
> These results confirm that RCPU scales effectively to larger LLMs.

---

> > ### Author Response · Authors · 2025-11-27
> > **[Weakness3, Question3] Clarification on generalization to unstructured or semi-structured pruning**
> >
> > The rotation scheme can be applied after unstructured or semi-structured pruning, but it does not directly generalize in a way that preserves sparsity.
> > Thus, additional constraints or a sparsity-preserving projection would possibly make the scheme compatible with unstructured or semi-structured sparsity patterns.
> > We consider this an interesting direction for future work.

---

> ### Author Response · Authors · 2025-11-26
> **[Minor comments, Questions4] typo fixed**
>
> > (“an rotation-constrained” → “a rotation-constrained”)
>
> We appreciate the comment. We fixed it in the revised manuscript.

---

> > ### Comment · Reviewer_f6Ag · 2025-11-27
> > **Official Comment by Reviewer f6Ag**
> >
> > Thanks for the authors’ replies. Questions 1 and 3 still have not been addressed.

---

> > > ### Author Response · Authors · 2025-11-27
> > >
> > > Thanks for the follow-up.
> > > We are currently addressing the remaining comments and would appreciate a bit more time.

---

> ### Author Response · Authors · 2025-11-27
> **Weakness1,Question1] Clarification on the relation to RotPruner and DenoiseRotator**
>
> Thank you for the helpful question. Clarifying the distinction from RotPruner and DenoiseRotator indeed makes the role of our approach clearer.
>
> While these methods learn orthogonal transformations to obtain pruning-friendly parameterizations before or during parameter removal,
> our method focuses on a different problem that arises after structured column pruning.
>
> In the related work section, we added the key difference from these methods to clarify the role of RCPU.
>
>
>
> > Rotation has also been used to improve prunability in other forms:
> RotPruner learns layer-wise orthogonal transforms to obtain pruning-friendly parameterizations,
> and DenoiseRotator trains rotations that concentrate importance scores before pruning.
> These methods use rotation as an additional parameterization to facilitate pruning before or during parameter removal.
> Unlike these rotation-learning or slicing-based approaches,
> RCPU focuses on reducing the pruning error after structured column removal through an analytically derived orthogonal compensation.

---

> > ### Comment · Reviewer_f6Ag · 2025-11-28
> > **Official Comment by Reviewer f6Ag**
> >
> > Thanks for the authors' replies. I would like to maintain my score and recommendation.

---

### Official Review · Reviewer_hZjV · 2025-10-22

**Soundness:** 2
**Presentation:** 3
**Contribution:** 1
**Rating:** 2
**Confidence:** 4

**Summary:**

The paper proposes a new method to compensate errors encountered during structured pruning of Large Language models. Generally their algorithm identifies columns (input channels) of the mlp_down projection (or attn_out) to prune and removes the corresponding structures (outputs channels) of the up projection.
Once the columns are identified they propose to update the remaining parameters to optimize the loss over a calibration dataset. Instead of updating all non-pruned parameters they only train a rotation matrix, their main motivation being to mitigate overfitting.

Their algorithm can be applied efficiently for each matrix in isolation, without requiring gradients or similar through the entire model.

They evaluate their method on Llama 7B and show tiny improvements over previous work "FLAPS" but their method also still has a 2% downstream accuracy drop even when only pruning 10% of the model weights.

**Strengths:**

The method is conceptually very simple and can be efficiently implemented and also in principle scaled to other methods. It only requires simple linear algebra. Furthermore, since it does structured pruning, the resulting model can readily be served and the speedups easily realized (as opposed to unstructured pruning for example).
The idea of constraining the updates to rotations seems interesting and new. Overall it is easy to follow the paper, but that's partially because it is not very deep.

**Weaknesses:**

- The paper motivates their constraint to rotations by "overfitting" however, there is no data presented that this actually happening in any way. Furthermore there are much more straightforward standard regularization approaches (for example Tikhonov) beyond constraining to rotations.
- The loss 3 decouples over output dimensions. it is unclear why it should not be best to just optimize each output channel independently.
- The evaluation is pretty constraint and only done on a single, quite outdated and rather small model (Llama-1 7B).
- The improvements over existing works are marginal (<0.5%), but the large accuracy gap to the dense model remains(2%-9%). Thus the practical relevance  of this work is very minor.

**Questions:**

- Can you provide empirical insights that the overfitting could not simply be mitigated by a simple ridge regression or similar?

---

> ### Author Response · Authors · 2025-11-26
> **[Weakness1 and Question1] Overfitting issues in ridge regression**
>
> Thank you for your constructive comments.
> Following reviewer hZjV's suggestion, we added a discussion on the potential overfitting problem when using Ridge regression, which further helps to clarify the strengths of our RCPU.
>
> We summarize the additional experiments as follows:
>
> - We compared RCPU with least-squares fitting under Ridge regularization and found that Ridge tends to overfit in our setting, highlighting the superior performance of RCPU.
> - The experimental findings are supported by PPL results, benchmark performance, and an analysis of the degrees of freedom.
>
> We have revised the manuscript to include the following discussion:
>
> To assess the extent to which RCPU is resistant to overfitting, we compare our RCPU against a regularized least-squares baseline using Ridge regression.
> Specifically, we consider the following compensation, which penalizes deviations from the original pruned weights $\mathbf{W_K}$:
>
> $\mathbf{W_K^\star} = \mathop{\arg \min}\limits_{\mathbf{W^\star} \in \mathbb{R}^{d_{\text{out}} \times d'}}(||\mathbf{Y-W^\star X_K||^2_F} + \lambda ||\mathbf{W^\star - W_K}||^2_F)$
>
> Closed form is:
>
> $
> \mathbf{W}_K^\star
> = (\mathbf{Y}\mathbf{X}_K^\top + \lambda \mathbf{W}_K)
> (\mathbf{X}_K \mathbf{X}_K^\top + \lambda \mathbf{I})^{-1}
> $
>
> In this equation, the weights can be adjusted, but the penalty keeps them close to the original. We performed a grid search over $\lambda \in \{10^{-6}, 10^{-5}, \ldots, 10^{6} \}$, and reported the result on best $\lambda$.
> [Figure 3](https://anonymous.4open.science/r/anonymous-llm-pruning-D884/figure3.png) focuses on the effect of different error-compensation methods (LS or Rot.) in different calibration sizes.
> From the figures of (a) and (b), we observe that with
> $N_\mathrm{calib} = 128$, the proposed rotation-based compensation achieves the best PPL. In (a), the regularized LS even worsens PPL, reflecting its tendency to overfit under limited calibration.
> In contrast, the figures of (c) and (d) show that when
> $N_\mathrm{calib} = 512$, both rotation and Ridge-based compensation effectively reduce PPL, but the regularized LS update contributes more strongly to the improvement.
> Although it is intuitively expected that the effect of the least square fitting becomes larger as the number of calibration samples increases, we emphasize that this does not necessarily translate into better downstream benchmark performance.
> Indeed, for example in [Table 6](https://anonymous.4open.science/r/anonymous-llm-pruning-D884/table6.png), LS-based compensations are not the top performer, whereas RCPU often achieves the best accuracy.
> Moreover, we discuss overfitting issues in LS+Ridge from the viewpoint of the degree of freedom. According to [The elements of statistical learning](https://www.sas.upenn.edu/~fdiebold/NoHesitations/BookAdvanced.pdf), the degree of freedom in ridge regression is computed by $d_\mathrm{out} \sum_i \frac{\sigma_{i}^2}{\sigma_{i}^2 + \lambda}$, where $\sigma$ denotes the singular value of the input matrix.
> Using this equation and the best $\lambda$, we obtain the degree of freedom values in the range $1.395 \times 10^9$ to $1.578 \times 10^9$ across each pruning ratio.
> In contrast, the degree of freedom in RCPU is given by $\frac{d_\mathrm{out}(d_\mathrm{out}-1)}{2}$ since $Q$ is constrained to the orthogonal matrix. And its value is $5.36 \times 10^8$, which is smaller than that of LS+Ridge.
> This indicates that, from the standpoint of preserving the pretrained knowledge of LLMs, the rotation-based compensation tends to be more robust.
> We also highlight that, in the context of LLMs, selecting an appropriate regularization hyper-parameter $\lambda$ can be computationally expensive, as it requires repeatedly computing large matrix inverses for multiple candidate values of $\lambda$.
> In contrast, our method has no hyper-parameters, avoiding this overhead.

---

> > ### Comment · Reviewer_hZjV · 2025-11-27
> > **Re: [Weakness1 and Question1] Overfitting issues in ridge regression**
> >
> > Thank you very much for the additional experiments with Ridge Regression and updating the paper with the theory and data. It's quite interesting that with 512 calibration samples RCPU does quite consistently worse on perplexity, but better in terms of downstream accuracy (at least on the considered two Llama models).
> >
> > Could you please also generate a plot with the perplexity as a function of lambda when running LS for both models / calibration sizes? I appreciate that you compare against the best one, but it is additionally interesting to understand the behavior with varying regularization strengths.
> >
> > I don't think the degree of freedom argument alone drives the point home fully, as with large regularization we can also make that number very small.
> >
> > I rather think that (if anything) the inductive bias of constraining to rotations is one that is particularly suited to the investigated LLMs.

---

> > > ### Author Response · Authors · 2025-12-02
> > > **Re:Re: [Weakness1 and Question1] Overfitting issues in ridge regression**
> > >
> > > Thank you for your response. We plotted PPL versus $\lambda$ in [Figure5 (Llama-7B)](https://anonymous.4open.science/r/anonymous-llm-pruning-D884/figure5.png) and [Figure6 (Llama-2-13B)](https://anonymous.4open.science/r/anonymous-llm-pruning-D884/figure6.png).
> > > Overall, the best $\lambda$ varies widely depending on the model type, pruning ratio, and the number of calibration samples. This suggests that finding an optimal $\lambda$ in a reliable manner is inherently difficult.
> > > For Llama-7B, the best $\lambda$ chosen on the calibration set does not yield the best test-set performance, indicating insufficient generalization.
> > > For Llama-2-13B, the PPL becomes unstable for certain $\lambda$ values (especially near zero). This instability is particularly pronounced when the calibration size is 128, which likely reflects the severe mismatch between the number of parameters and the amount of available calibration data. In Llama-2-13B, the best $\lambda$ achieves performance comparable to RCPU when using 128 samples, and slightly better than RCPU when using 512 samples.
> > >
> > > However, across models and calibration sizes, RCPU consistently outperforms LS+Ridge on downstream tasks ([Table2](https://anonymous.4open.science/r/anonymous-llm-pruning-D884/table2.png), [Table3](https://anonymous.4open.science/r/anonymous-llm-pruning-D884/table3.png), [Table6](https://anonymous.4open.science/r/anonymous-llm-pruning-D884/table6.png) and [Table7](https://anonymous.4open.science/r/anonymous-llm-pruning-D884/table7.png)). This indicates that LS+Ridge compensation achieves some degree of in-domain generalization but fails to generalize out-of-domain. From the perspective of preserving the pretrained knowledge of the LLM, RCPU provides a more robust form of compensation.

---

> ### Author Response · Authors · 2025-11-26
> **[Weakness3] Additional model evaluation**
>
> Thank you for the constructive comments.
>
> To evaluate generality beyond Llama-7B, we additionally tested RCPU on Llama-2-13B using PPL and benchmark tasks.
> As shown in [Figure4](https://anonymous.4open.science/r/anonymous-llm-pruning-D884/figure4.png), [Table1](https://anonymous.4open.science/r/anonymous-llm-pruning-D884/table1.png), [Table3](https://anonymous.4open.science/r/anonymous-llm-pruning-D884/table3.png) and [Table7](https://anonymous.4open.science/r/anonymous-llm-pruning-D884/table7.png), RCPU maintains superior performance and follows the same trend as on Llama-7B, while absolute scores increase due to the stronger capability of the larger model.
> These results confirm that RCPU scales effectively to larger LLMs.

---

> > ### Comment · Reviewer_hZjV · 2025-11-27
> > **Re: [Weakness3] Additional model evaluation**
> >
> > Thank you for considering Llama-2 13B additionally.
> > I think the work needs even more models. Your work is quite empirically motivated (RCPU better than LS in downstream performance, but not in perplexity) without clearly convincing arguments that this generalizes.
> > To be more explicit: my wishlist would be:
> > - Llama-3.3 70B Instruct
> > - some Mistral model
> > - some Qwen3 model

---

> > > ### Author Response · Authors · 2025-12-02
> > > **Re:Re: [Weakness3] Additional model evaluation**
> > >
> > > Thank you for your constructive response.
> > > Due to an implementation issue, we were not able to immediately test the models you suggested in the wishlist.
> > > Instead, we conducted evaluations on Vicuna-7B and obtained the results shown in [Table8 (PPL)](https://anonymous.4open.science/r/anonymous-llm-pruning-D884/table8.png), [Table9(Benchmark 128)](https://anonymous.4open.science/r/anonymous-llm-pruning-D884/table9.png), and [Table10(Benchmark 512)](https://anonymous.4open.science/r/anonymous-llm-pruning-D884/table10.png).
> > > Both the perplexity and benchmark evaluations followed almost similar trends to those observed in other models, although the magnitude of the improvement varied.

---

> ### Author Response · Authors · 2025-11-26
> **[Weakness4] Discussion of RCPU's improvement**
>
> Structured pruning is known to incur larger performance degradation relative to dense models, particularly when compared with distillation-based compression techniques.
> This is a known challenge for the structured pruning community as a whole.
> While distillation can be effective in recovering accuracy, it often requires substantial computational resources and a full training pipeline, which may only be available to a limited number of organizations.
> For this reason, exploring alternative approaches like structured pruning is valuable.
> In particular, the baselines and our method are designed for settings with limited calibration data, making them more broadly applicable in practice.
> In this context, SliceGPT is a famous and known as an effective structured pruning method.
> In [Table3](https://anonymous.4open.science/r/anonymous-llm-pruning-D884/table3.png), we added the comparison with SliceGPT.
> Our results show that RCPU exceeds SliceGPT by 3–9\% in mean accuracy, which is a considerable gap relative to typical gains reported in this area.
> For this reason, we believe the improvements achieved by RCPU are not minor.

---

> ### Author Response · Authors · 2025-11-27
> **[Weakness2] On Output-Dimension Decoupling**
>
> Thank you for the comment. As you pointed out, the loss
>
> $\mathcal{L}(\mathbf{A}) = \|| \mathbf{Y - A Z}\||_F^2$
>
> can be written in a way that allows each row (output dimension) to be optimized separately.
> This is also true for the ridge-regularized version,
>
> $\mathbf{W_K^\star} = \mathop{\arg \min}\limits_{\mathbf{W^\star} \in \mathbb{R}^{d_{\text{out}} \times d'}}(||\mathbf{Y-W^\star X_K||^2_F} + \lambda ||\mathbf{W^\star - W_K}||^2_F).$
>
> ~~However, we do not use such row-by-row optimization in this work.~~
> ~~The model considered in our setting is a Transformer, and its output dimensions are not independent of each other.~~
> ~~Each output vector is formed through attention and feed-forward layers, and the dimensions are considered to be correlated.~~
> ~~If each row were optimized separately, these relationships could be distorted.~~
>
> ~~For this reason, we optimize the entire matrix at once, rather than treating each row independently.~~

---

> > ### Comment · Reviewer_hZjV · 2025-11-27
> > **Re: [Weakness2] On Output-Dimension Decoupling**
> >
> > I am a bit confused. For Ridge regression the output channels are fully decoupled. And that is the objective you optimize. This optimization has no awareness at all that it is within a Transformer model or similar. So in my opinion there is no coupling. Whether algorithmically you solve each row separately or not does not change that fact (unless you choose to optimize $\lambda$ for each row, which would be an overkill).
> >
> > So in particular I think the following point in Section 2.2 is wrong.
> >
> > > (iii) Undesired coupling: Unconstrained updates
> > based on least-square fitting modify all output dimensions simultaneously, potentially degrading
> > well-preserved directions to correct poorly preserved ones. Desideratum: prefer updates that act
> > locally with respect to the kept subspace and maintain structure across output directions.
> >
> >
> > My intuition is rather that RCPU *couples* the output dimensions in a way that benefits downstream performance. I could for example imagine that it is more robust to outliers that have not been present in the calibration data or sth similar.

---

> > > ### Author Response · Authors · 2025-11-28
> > > **Re:Re: [Weakness2] On Output-Dimension Decoupling**
> > >
> > > We confirmed what the reviewer hZjV pointed out in the previous comments was correct.
> > > The equation
> > >
> > > $\mathcal{L}(\mathbf{A}) = \| \mathbf{Y - A Z}\|_F^2$
> > >
> > > was decoupled.
> > > As you pointed out, our intention is to couple output dimensions effectively.
> > > To compare the robustness against overfitting, we replaced the former baseline based on
> > >
> > > $\mathcal{L}(\mathbf{A}) = \| \mathbf{Y - A Z}\|_F^2$
> > >
> > > with the ridge baseline based on
> > >
> > > $\mathcal{L}(\mathbf{W^\star}) = ||\mathbf{Y-W^\star X_K||^2_F} + \lambda ||\mathbf{W^\star - W_K}||^2_F$.
> > >
> > > We have replaced all results with those based on the latter equation.
> > > We have removed the claim (iii) because it was ambiguous description.
> > >
> > > We appreciate the reviewer's contribution to clarification of the paper contents.
> > > We remove the misleading points in our previous comments.

---

### Official Review · Reviewer_hBF1 · 2025-10-30

**Soundness:** 3
**Presentation:** 3
**Contribution:** 2
**Rating:** 6
**Confidence:** 2

**Summary:**

The paper proposes **RCPU (Rotation-Constrained Parameter Update)**, a structured pruning method designed to minimize post-pruning output distortion in large language models. After column-wise pruning, the authors introduce an **orthogonal Procrustes-based reparameterization** to align the preserved subspace with the original outputs, optionally scaled by a global scalar factor. This design preserves vector norms and inner products, aiming to maintain geometric consistency with minimal calibration data. Additionally, the paper presents a variance-aware column importance score
 to prioritize dimensions contributing more significantly to dominant output directions.
 Experiments on LLaMA-7B (using 128 WikiText-2 calibration samples) show that RCPU outperforms or matches WANDA-sp and FLAP in perplexity and several zero-shot downstream benchmarks.

**Strengths:**

1. **Clean and Intuitive Formulation**
   The method constrains post-pruning compensation to an orthogonal transformation, which is both mathematically sound and conceptually clear. The use of Procrustes alignment offers a closed-form SVD solution that preserves geometric structure and mitigates overfitting.

2. **Innovative Use of Rotation Constraints**
    Introducing rotation constraints to reduce pruning error is a novel and effective idea. It maintains representational geometry while minimizing reconstruction error without retraining.

3. **Layer-Agnostic and Easily Transferable**
    The approach is layer-wise, gradient-free, and architecture-agnostic, allowing easy integration into existing pruning pipelines and adaptation to other large models.

**Weaknesses:**

1. **Limited Experiments and Comparisons**
   The experiments were conducted only on the 7B model, with no evaluation on larger models. Analysis over varying calibration sizes (8, 32, 64, 128, 512) is needed for robustness. Additionally, comparisons are limited to FLAP and WANDA-sp, ignoring other state-of-the-art methods.
2. **Unclear Strengths and Limitations of Rotation Constraint**
   While the paper claims orthogonal alignment preserves inner products and mitigates overfitting, it lacks formal analysis. Additionally, the reasons for not considering other constraints, like regularized least squares, are not discussed.

**Questions:**

1. The anonymized code repository appears incomplete; most files are missing except for *main.py*. Could the authors verify that the full implementation has been properly uploaded?
2. Did the authors visualize how the scores $\gamma_j$ are distributed across columns?
3. What's the choice of scaled variant in your experiments, Please clarify how this choice affects the reported results ?

---

> ### Author Response · Authors · 2025-11-27
> **[Weakness1, Weakness2] Additional experiments**
>
> Thank you for your constructive comments. As the reviewer hBF1 pointed out, we can further improve our paper quality by adding the model variety, comparison with other method and the evaluation on regularized least square method.
> As summary, we showed the following:
>
> - We added experiments on Llama-2-13B, which is larger and newer one. We confirmed the similar tendency to Llama-7B.
> - We added the comparison with other state of the art method, SliceGPT. Our method outperformed SliceGPT in many cases.
> - We added the evaluation on the different number of calibration sets.
> - We compared RCPU with the least square fitting with Ridge regularizer. We showed that the least square with Ridge tends to overfit in our settings and superiority of our RCPU. The experimental result is based on PPL results, benchmark resutls, and the analysis of the degree of freedom.
>
> We revised the manuscript based on the following explanation, which include all the requirements.
>
> First of all, we conducted experiments regarding PPL, which changes $N_{\mathrm{calib}}$ (The number of calibration samples). [Figure2](https://anonymous.4open.science/r/anonymous-llm-pruning-D884/figure2.png) reports how the perplexity of RCPU varies with the number of calibration samples.
> We observe that PPL drops rapidly as $N_{\mathrm{calib}}$ increases and becomes roughly stable once $N_{\mathrm{calib}}$ reaches around 64. Based on this trend, we adopt $N_{\mathrm{calib}}=$ 128 and 512 as the calibration-set sizes for other experiments in this study.
> While 128 is a common choice in prior work, we additionally include 512, which lies well within the empirically stable region, providing more reliable evaluations.
>
> Due to space limitation, we separately describe (i)"Overfitting mitigation and comparison with the regularization method", (ii) "Larger model evaluation and comparison with other SoTA method"

---

> ### Author Response · Authors · 2025-11-27
> **(i) Overfitting mitigation and comparison with the regularization method**
>
> To assess the extent to which RCPU is resistant to overfitting, we compare our RCPU against a regularized least-squares baseline using Ridge regression.
> Specifically, we consider the following compensation, which penalizes deviations from the original pruned weights $\mathbf{W_K}$:
>
> $\mathbf{W_K^\star} = \mathop{\arg \min}\limits_{\mathbf{W^\star} \in \mathbb{R}^{d_{\text{out}} \times d'}}(||\mathbf{Y-W^\star X_K||^2_F} + \lambda ||\mathbf{W^\star - W_K}||^2_F)$
>
> Closed form is:
> $
> \mathbf{W}_K^\star
> = (\mathbf{Y}\mathbf{X}_K^\top + \lambda \mathbf{W}_K)
> (\mathbf{X}_K \mathbf{X}_K^\top + \lambda \mathbf{I})^{-1}
> $
>
> In this equation, the weights can be adjusted, but the penalty keeps them close to the original. We performed a grid search over $\lambda \in \{10^{-6}, 10^{-5}, \ldots, 10^{6}\}$, and reported the result on best $\lambda$.
> [Figure 3](https://anonymous.4open.science/r/anonymous-llm-pruning-D884/figure3.png) focuses on the effect of different error-compensation methods (LS or Rot.) in different calibration sizes.
> From the figures of (a) and (b), we observe that with $N_\mathrm{calib} = 128$, the proposed rotation-based compensation achieves the best PPL.
> In (a), the regularized LS even worsens PPL, reflecting its tendency to overfit under limited calibration.
> In contrast, the figures of (c) and (d) show that when $N_\mathrm{calib} = 512$, both rotation and Ridge-based compensation effectively reduce PPL, but the regularized LS update contributes more strongly to the improvement.
> Although it is intuitively expected that the effect of the least square fitting becomes larger as the number of calibration samples increases, we emphasize that this does not necessarily translate into better downstream benchmark performance.
> Indeed, for example in [Table 6](https://anonymous.4open.science/r/anonymous-llm-pruning-D884/table6.png), LS-based compensations are not the top performer, whereas RCPU often achieves the best accuracy.
> Moreover, we discuss overfitting issues in LS+Ridge from the viewpoint of the degree of freedom. According to [The elements of statistical learning](https://www.sas.upenn.edu/~fdiebold/NoHesitations/BookAdvanced.pdf), the degree of freedom in ridge regression is computed by $d_\mathrm{out} \sum_i \frac{\sigma_{i}^2}{\sigma_{i}^2 + \lambda}$, where $\sigma$ denotes the singular value of the input matrix.
> Using this equation and the best $\lambda$, we obtain the degree of freedom values in the range $1.395 \times 10^9$ to $1.578 \times 10^9$ across each pruning ratio.
> In contrast, the degree of freedom in RCPU is given by $\frac{d_\mathrm{out}(d_\mathrm{out}-1)}{2}$ since $Q$ is constrained to the orthogonal matrix. And its value is $5.36 \times 10^8$, which is smaller than that of LS+Ridge.
> This indicates that, from the standpoint of preserving the pretrained knowledge of LLMs, the rotation-based compensation tends to be more robust.
> We also highlight that, in the context of LLMs, selecting an appropriate regularization hyper-parameter $\lambda$ can be computationally expensive, as it requires repeatedly computing large matrix inverses for multiple candidate values of $\lambda$.
> In contrast, our method has no hyper-parameters, avoiding this overhead.

---

> ### Author Response · Authors · 2025-11-27
> **(ii) Larger model evaluation and comparison with other SoTA method**
>
> ## Larger model evaluation
> To evaluate generality beyond Llama-7B, we additionally tested RCPU on Llama-2-13B using PPL and benchmark tasks.
> As shown in [Figure4](https://anonymous.4open.science/r/anonymous-llm-pruning-D884/figure4.png), [Table1](https://anonymous.4open.science/r/anonymous-llm-pruning-D884/table1.png), [Table3](https://anonymous.4open.science/r/anonymous-llm-pruning-D884/table3.png) and [Table7](https://anonymous.4open.science/r/anonymous-llm-pruning-D884/table7.png), RCPU maintains superior performance and follows the same trend as on Llama-7B, while absolute scores increase due to the stronger capability of the larger model.
> These results confirm that RCPU scales effectively to larger LLMs.
> ## Comparison with SliceGPT
> We choose a new state of the art baseline, SliceGPT, which we have mentioned in the related work section.
>
> Since SliceGPT does not support Llama-1-7B, we conducted the performance comparison on Llama-2-13B. [Table3](https://anonymous.4open.science/r/anonymous-llm-pruning-D884/table3.png) and [Table7](https://anonymous.4open.science/r/anonymous-llm-pruning-D884/table7.png) show the benchmark results on Llama-2-13B in $N_\mathrm{calib}=$128 and 512.
> SliceGPT transforms the entire model into an equivalent structure using an orthogonal matrices, and then performs row or column deletion in a single global step.
> Due to this property of applying a global transformation followed by global pruning, we expect that, at high pruning ratios, a mismatch arises between the input distributions assumed by each layer and the actual input distributions after pruning.
> This mismatch is also expected to accumulate across layers and thus we believe accuracy tends to degrade at high pruning ratios.
> In contrast, RCPU optimizes the pruning-induced error layer by layer.
> Thus, even at high pruning ratios, each layer is more likely to maintain representations close to the inputs it assumes, which we believe leads to better benchmark performance.

---

> ### Author Response · Authors · 2025-11-27
> **[Question 1] Regarding the code repository**
>
> Thank you for pointing this out.
> We double-checked the anonymized repository, and all files are indeed included in the uploaded package. It seems that GitHub’s anonymous view occasionally fails to display the full directory on first access.
>
> To ensure that reviewers can reliably access the complete implementation, we refreshed the archive and re-uploaded the repository.
> The link remains the same, and the full codebase should now be visible without issues.
>
> Please let us know if any specific file still appears missing.

---

> ### Author Response · Authors · 2025-11-27
> **[Question3] Scaled variant choice**
>
> The ``scaled variant'' used in our experiments is exactly the one described at the end of Section 3.1 (Equation (9)).
> When pruning removes a subset of columns, the output norm of the pruned sublayer tends to shrink.
> To mitigate this, we apply a single global scalar $s^\star$ jointly with the rotation $Q^\star$, as defined in the paper.
> This is the only scaling scheme used in all reported results.
>
> In addition, we conducted further experiments with a larger calibration set (512 samples instead of 128).
> Comparing [Table3(512)](https://anonymous.4open.science/r/anonymous-llm-pruning-D884/table3.png) with [Table7(128)](https://anonymous.4open.science/r/anonymous-llm-pruning-D884/table7.png), the scaled variant performed better and ranked as the best baseline more often in the 512-sample setting than in the 128-sample setting.
> Intuitively, the additional samples stabilize the norm statistics of the unpruned versus pruned outputs, allowing the global scale $s^\star$ to more effectively restore the original magnitude.

---

### Official Review · Reviewer_SQAA · 2025-11-02

**Soundness:** 3
**Presentation:** 3
**Contribution:** 3
**Rating:** 6
**Confidence:** 4

**Summary:**

The paper proposes RCPU, a post‑pruning compensation method for LLMs that restricts the update to an orthogonal rotation (optionally with a single global scale) computed via an Orthogonal Procrustes fit between the pruned layer’s outputs and the original outputs on a small calibration set. This preserves output norms and inner products, reducing overfitting and geometric distortion that plague unconstrained least‑squares fixes. It’s paired with a simple variance‑aware column score, (\gamma_j=|W_{:,j}||X_{j,:}|\operatorname{Var}(X_{j,:})), to keep columns likely to support principal output directions. Applied layerwise to LLaMA‑7B with 128 WikiText‑2 calibration samples, RCPU consistently lowers perplexity and improves or matches zero‑shot accuracy versus WANDA‑sp and the bias‑only FLAP baseline across 10–30% structured pruning. Figure 1 (p.2) sketches the pipeline; Algorithm 1 (p.5) gives the exact steps; Tables 1–2 (pp.6–7) report the gains; Tables 3–4 (p.8) show where to apply it and the runtime/memory trade‑offs.

Key contributions

* Rotation‑constrained compensation: Formulates post‑pruning recovery as a closed‑form Orthogonal Procrustes problem and updates only the retained columns via (W_K \leftarrow Q^\star W_K) (optionally (s^\star Q^\star W_K)), preserving geometry while aligning to the original outputs (Sec. 3.1, Fig. 1).
* Variance‑aware selection rule: Extends WANDA‑sp’s magnitude/activation score with an input‑variance term to prefer columns supporting dominant directions, improving compatibility with the rotation update (Sec. 3.2).
* Drop‑in, no finetuning, modest cost: Layerwise, closed‑form SVD per treated sublayer; no architectural changes; measured pruning time under ~10 s per layer on LLaMA‑7B while reducing parameters and memory with higher pruning ratios (Algorithm 1; Table 4).
* Empirical gains on LLaMA‑7B: Lower perplexity than WANDA‑sp and typically better than FLAP at 10–30% pruning (e.g., at 20%: PPL 14.40 vs 16.70/15.36) and competitive zero‑shot accuracy across BoolQ, PIQA, HellaSwag, WinoGrande, ARC‑e/c, OBQA; rotation outperforms unconstrained least‑squares fixes, especially as pruning increases (Tables 1–2; Fig. 2).
* Ablations on where to rotate: Best results when compensating both attention (o_proj) and MLP (down_proj); (o_proj) alone helps more than (down_proj) alone (Table 3).

**Strengths:**

Strengths: originality

* Recasts post‑pruning error correction as an orthogonal Procrustes alignment on the layer outputs, restricting the compensation to a rotation (optionally with a single global scale). This is a crisp, geometry‑preserving reframing that removes a well‑known limitation of unconstrained least‑squares fixes under tiny calibration sets: shear/scale distortions and overfitting. The formulation and closed‑form solution are given in Sec. 3.1, with the pipeline visualized in Figure 1 (p. 2).
* Pairs that rotation with a variance‑aware column score that augments magnitude and activation scale with input variance, which is a simple but thoughtful twist that prefers columns supporting principal output directions (Sec. 3.2). The idea is original in the structured‑pruning context and complementary to existing heuristics such as Wanda‑sp.
* Offers a drop‑in procedure that can be inserted immediately after column pruning without architectural changes or fine‑tuning, expressed succinctly in Algorithm 1 (p. 5). The combination of a classic SVD‑based alignment with modern LLM pruning is a creative, low‑friction synthesis.

Strengths: quality

* Methodological soundness. The paper motivates the failure modes of unconstrained least squares, formalizes the rotation‑constrained problem, and provides a closed‑form optimizer via SVD along with a scaled variant. The constraints directly preserve norms and inner products, aligning with the desiderata stated in Sec. 2.2 and solved in Sec. 3.1. Complexity is analyzed (O(d_out³) per treated sublayer), and the computational steps are transparent (Sec. 3.3).
* Strong empirical evidence. On LLaMA‑7B with only 128 WikiText‑2 calibration samples, rotation consistently reduces perplexity vs. Wanda‑sp and typically matches or beats FLAP across 10–30% pruning. Table 1 (p. 6) shows, for example, at 20% pruning: PPL 14.40 (RCPU Rot.) vs. 16.70 (Wanda‑sp) and 15.36 (FLAP). Figure 2 (p. 7) further demonstrates that least‑squares compensation often harms PPL while rotation reliably helps, under both scoring rules.
* Breadth of evaluation. Zero‑shot accuracy is reported on seven diverse language understanding tasks; RCPU variants are competitive and often superior to baselines, including at higher pruning where accuracy is most fragile (Table 2, p. 7). The study also includes targeted ablations that identify where compensation matters most: updating both attention o_proj and MLP down_proj yields the largest gains (Table 3, p. 8).
* Practicality demonstrated. The efficiency table reports per‑layer pruning times under 10 s and monotonic memory savings with higher pruning ratios (Table 4, p. 8), strengthening the claim that the method is deployable without exotic infrastructure.

Strengths: clarity

* The paper is exceedingly readable: notation is introduced cleanly (Sec. 2.1), the failure modes of LS are enumerated before the proposed fix (Sec. 2.2), and the solution path is linear. Figure 1 on page 2 gives an at‑a‑glance workflow that matches Algorithm 1 on page 5, which makes the procedure easy to reimplement.
* Tables and plots are well‑chosen and interpretable: Table 1 isolates PPL vs. pruning ratio; Figure 2 explicitly contrasts “no compensation,” “LS,” and “RCPU” under both scoring rules; Table 2 summarizes per‑task and mean accuracies; Tables 3–4 provide ablations and resource figures. The accompanying text consistently ties these visuals back to the stated hypotheses.
* Reproducibility details are reasonable: calibration dataset, number of samples, layers targeted, evaluation harness, and hardware are all specified (Sec. 4.1), and an anonymized code link is provided.

Strengths: significance

* Practical impact for structured pruning. Many deployments can only spare tiny calibration sets and cannot fine‑tune; a one‑shot, geometry‑preserving compensation that is drop‑in and SVD‑simple is immediately useful. Gains are most pronounced exactly where practitioners need them: at higher pruning ratios (e.g., 30% PPL improvements over Wanda‑sp in Table 1), where retaining performance is hardest.
* Conceptual influence. Framing error recovery as orientation correction rather than generic regression is a clean lens that could inform future compression methods, e.g., combining with other structure‑preserving transforms or smarter selection rules. The ablation showing o_proj benefits more than down_proj (Table 3) offers actionable insights about where geometry matters most in transformer blocks.
* Broad applicability. Because the update is layer‑local, closed‑form, and architecture‑agnostic, it should transport to other LLM families and even beyond LMs to any setting where structured pruning removes subspaces and small calibration sets are the norm. The efficiency numbers and memory reductions in Table 4 make this a credible path for edge or resource‑constrained deployments.

Overall: a neat, principled, and practical piece of work. The rotation‑constrained view is conceptually tidy, the experiments are careful and targeted, and the clarity of exposition makes the method easy to adopt.

**Weaknesses:**

* The central idea is to restrict post‑pruning compensation to an orthogonal Procrustes fit. That’s a neat reuse of a classic tool, but the paper underplays overlap with closely related “structure‑preserving reparameterizations” such as SliceGPT, which also leverages orthogonal transforms to manage structured deletions. The Related Work section asserts a difference but offers no head‑to‑head comparison or synthetic study clarifying when layer‑local Procrustes is preferable to a single global reparameterization. Add a direct comparison against SliceGPT, both as a baseline and as a combined pipeline (SliceGPT first, then RCPU), and make the distinctions operational rather than rhetorical. Cite Figure 1 for the RCPU pipeline and §5.3 for the SliceGPT discussion, then show empirical deltas.

* “Geometry preserving” is asserted, not quantified. Orthogonal maps preserve norms and pairwise angles within the calibration subspace, but the paper does not measure whether downstream layers see closer activations after compensation. Add layerwise geometric diagnostics: cosine similarity and norm‑ratio histograms between original outputs Y and compensated outputs QZ at calibration and test time; report mean and tail statistics per sublayer. Tie these to PPL changes to show that geometry preservation correlates with generalization. Anchor this to the optimization in §3.1 and the workflow in Figure 1.

* The choice to rotate across the full dout mixes attention heads in o_proj. That may be unnecessary coupling. Test block‑diagonal Q that respects head boundaries vs full Q, and report both quality and compute trade‑offs. The ablation in Table 3 shows where rotation helps (o_proj vs down_proj), but not whether cross‑head mixing is essential. This is low‑effort to implement and could improve robustness.

* The scaled variant uses a single global s; Table 1 shows it is largely neutral. Before discarding scaling entirely, probe slightly richer but still well‑conditioned classes: per‑channel diagonal scaling or a small number of Householder reflections. These preserve most geometry while giving capacity to fix systematic magnitude shifts that a single s cannot capture. Evaluate them beside “Rot.” and “Rot.+Scale.” Refer to §3.1–3.3 and Table 1.

* Dependence on calibration size and distribution is not analyzed. All results use 128 WikiText‑2 samples. Provide calibration‑size curves (e.g., N ∈ {32, 64, 128, 512}) and a distribution‑shift test: calibrate on WikiText‑2 but evaluate on the seven downstream tasks, and vice versa. Figure 2 already contrasts LS vs RCPU; extend that figure style to show RCPU’s sample‑efficiency and brittleness under shift. This directly tests the “less prone to overfitting” claim from §2.2–§3.1.

* The variance‑aware score is only partially ablated. Figure 2 contrasts WANDA‑sp vs the proposed score, but the score itself bundles three factors. Add controlled ablations: remove just the variance term, replace sample variance with a shrinkage estimator, and test robust alternatives (median absolute deviation). Report how each affects M = YZ⊤ conditioning and the singular value spectrum that drives Q. Ground this in §3.2 and Figure 2.

* Numerical details of Procrustes are missing. In practice, reflections (det(Q) < 0) can arise; some implementations flip the last column of U to enforce det(Q) = 1. State which variant you use and whether it matters for quality. Also note precision (fp16 vs fp32) and any stabilization for tall‑skinny Z multiplication. This belongs in §3.1–3.3.

**Questions:**

1. “Geometry preservation” beyond intuition.
   The paper asserts that rotations preserve norms and inner products and reduce distortion relative to LS (§2.2, §3.1), but this is not measured directly.
   Request: Report cosine similarity and norm‑ratio histograms between original outputs Y and compensated outputs QZ on held‑out data. Correlate those with perplexity deltas per sublayer. This would substantiate the geometric motivation. (Fig. 1; §2.2–§3.1)

2. Calibration size and distribution shift.
   All results use 128 WikiText‑2 samples (§4.1). The stated advantage is reduced overfitting under small calibration, but sample‑efficiency is not quantified.
   Request: Provide calibration‑size curves (N ∈ {32, 64, 128, 512}) and a distribution‑shift test: calibrate on WikiText‑2, evaluate on the seven tasks, and vice versa. Present curves similar to Figure 2 to show robustness. (Fig. 2; §4.1)

3. When not to rotate.
   Practitioners need a diagnostic to decide whether a sublayer will benefit.
   Request: Report a simple predictor such as the principal angle between span(Z) and span(Y) or the condition number of M = YZ⊤, and plot expected PPL change vs that quantity. This could become a decision rule for where to apply RCPU. (Eq. 6; Table 3)

---

> ### Author Response · Authors · 2025-11-27
> **Thank you for your review**
>
> We appreciate reviewer SQAA’s detailed assessment and the many suggestions for deeper analysis.
> Given the time constraints of the rebuttal period, we prioritized addressing (i) points raised by multiple reviewers and (ii) issues that most directly affect the paper’s clarity and claims.
> In the following, we first respond to the concerns that overlap with other reviewers, and then address the remaining points to the extent possible.

---

> ### Author Response · Authors · 2025-11-27
> **[Weakness4] Regarding a single global s**
>
> Thank you for your constructive advice about possible effectiveness of the scaling variant.
> Inspired by your advice, we thought that the scaling variant might be effective when the number of calibration sets increases.  We conducted further experiments with a larger calibration set (512 samples instead of 128) to ensure the effectiveness of scaling.
> The results are shown in [Table3(512)](https://anonymous.4open.science/r/anonymous-llm-pruning-D884/table3.png) and [Table7(128)](https://anonymous.4open.science/r/anonymous-llm-pruning-D884/table7.png).
>
> As we expected, the scaled variant performed better and ranked as the best baseline more often in the 512-sample setting than in the 128-sample setting.
> Intuitively, the additional samples stabilize the norm statistics of the unpruned versus pruned outputs, allowing the global scale $s^\star$ to more effectively restore the original magnitude.

---

> ### Author Response · Authors · 2025-11-27
> **[Weakness5,Question2] Influence of the calibration sample size and the distribution shift**
>
> To determine the number of calibration samples we use in the following experiments, first of all, we conducted experiments regarding PPL, which change $N_{\mathrm{calib}}$ (The number of calibration samples).
> [Figure2](https://anonymous.4open.science/r/anonymous-llm-pruning-D884/figure2.png) reports how the perplexity of RCPU varies with the number of calibration samples.
> We observe that PPL drops rapidly as $N_{\mathrm{calib}}$ increases and becomes roughly stable once $N_{\mathrm{calib}}$ reaches around 64. Based on this trend, we adopt $N_{\mathrm{calib}} = 128$ and 512 as the calibration-set sizes for other experiments in this study.
> While 128 is a common choice in prior work, we additionally include 512, which lies well within the empirically stable region, providing more reliable evaluations.
>
> To show RCPU's robustness to the distribution shift, we compare our RCPU against a regularized least-squares baseline using Ridge regression.
> Specifically, we consider the following compensation, which penalizes deviations from the original pruned weights $\mathbf{W_K}$:
>
> $\mathbf{W_K^\star} = \mathop{\arg \min}\limits_{\mathbf{W^\star} \in \mathbb{R}^{d_{\text{out}} \times d'}}(||\mathbf{Y-W^\star X_K||^2_F} + \lambda ||\mathbf{W^\star - W_K}||^2_F)$
>
> Closed form is:
> $
> \mathbf{W}_K^\star
> = (\mathbf{Y}\mathbf{X}_K^\top + \lambda \mathbf{W}_K)
> (\mathbf{X}_K \mathbf{X}_K^\top + \lambda \mathbf{I})^{-1}
> $
>
> In this equation, the weights can be adjusted, but the penalty keeps them close to the original. We performed a grid search over $\lambda \in \{10^{-6}, 10^{-5}, \ldots, 10^{6}\}$, and reported the result on best $\lambda$.
> [Figure 3](https://anonymous.4open.science/r/anonymous-llm-pruning-D884/figure3.png) focuses on the effect of different error-compensation methods (LS or Rot.) in different calibration sizes.
> From the figures of (a) and (b), we observe that with $N_\mathrm{calib} = 128$, the proposed rotation-based compensation achieves the best PPL.
> In (a), the regularized LS even worsens PPL, reflecting its tendency to overfit under limited calibration.
> In contrast, the figures of (c) and (d) show that when $N_\mathrm{calib} = 512$, both rotation and Ridge-based compensation effectively reduce PPL, but the regularized LS update contributes more strongly to the improvement.
> Although it is intuitively expected that the effect of the least square fitting becomes larger as the number of calibration samples increases, we emphasize that this does not necessarily translate into better downstream benchmark performance.
> Indeed, for example in [Table 6](https://anonymous.4open.science/r/anonymous-llm-pruning-D884/table6.png), LS-based compensations are not the top performer, whereas RCPU often achieves the best accuracy.
> Moreover, we discuss overfitting issues in LS+Ridge from the viewpoint of the degree of freedom. According to [The elements of statistical learning](https://www.sas.upenn.edu/~fdiebold/NoHesitations/BookAdvanced.pdf), the degree of freedom in ridge regression is computed by $d_\mathrm{out} \sum_i \frac{\sigma_{i}^2}{\sigma_{i}^2 + \lambda}$, where $\sigma$ denotes the singular value of the input matrix.
> Using this equation and the best $\lambda$, we obtain the degree of freedom values in the range $1.395 \times 10^9$ to $1.578 \times 10^9$ across each pruning ratio.
> In contrast, the degree of freedom in RCPU is given by $\frac{d_\mathrm{out}(d_\mathrm{out}-1)}{2}$ since $Q$ is constrained to the orthogonal matrix. And its value is $5.36 \times 10^8$, which is smaller than that of LS+Ridge.
> This indicates that, from the standpoint of preserving the pretrained knowledge of LLMs, the rotation-based compensation tends to be more robust.
> We also highlight that, in the context of LLMs, selecting an appropriate regularization hyper-parameter $\lambda$ can be computationally expensive, as it requires repeatedly computing large matrix inverses for multiple candidate values of $\lambda$.
> In contrast, our method has no hyper-parameters, avoiding this overhead.

---

> ### Author Response · Authors · 2025-12-02
> **[Weakness1] SliceGPT comparison**
>
> Thank you for the suggestion. We additionally conducted performance comparisons on a larger Llama-2-13B with SliceGPT. [Table3](https://anonymous.4open.science/r/anonymous-llm-pruning-D884/table3.png) and [Table7](https://anonymous.4open.science/r/anonymous-llm-pruning-D884/table7.png) show the results.
> SliceGPT converts the entire model into an equivalent structure using orthogonal matrices and then performs row/column deletion in a single global step.
> While this approach enables efficient global pruning, its global transformation followed by global pruning can introduce a mismatch between the assumed and actual input distributions at each layer, especially under high pruning ratios.
> This mismatch accumulates across layers, which we believe explains the degradation in benchmark performance observed at high pruning levels.
> In contrast, RCPU compensates pruning-induced errors in a layer-wise manner, enabling each layer to preserve representations closer to the inputs it expects. As a result, our RCPU maintains more stable accuracy at high pruning ratios.
> Combining SliceGPT and RCPU is not experimentally pursued because the two methods target different stages and objectives, and SliceGPT alone showed limited improvements in our additional experiments.

---

> ### Author Response · Authors · 2025-12-02
> **[Weakness3] On the Necessity of Full-Space Rotation**
>
> Thanks for the suggestion. In standard Transformers, the output projection $W_o$ already mixes all attention heads through a dense linear map. After pruning, our method adjusts this mixing by replacing the pruned projection. In other words, $Q$ determines how the remaining head information are combined, consistent with the role that $W_o$ plays in the pretrained model. For this reason, we do not view the full rotation as introducing unnecessary coupling. We therefore focus on the full $Q$ and do not pursue the block-diagonal variant here.

---

> ### Author Response · Authors · 2025-12-02
> **[Weakness7] Numerical details of Procrustes**
>
> As the reviewer correctly pointed out, the closed-form Procrustes solution can produce an orthogonal matrix $Q$ whose determinant is negative, i.e., a reflection. In our implementation, we simply use the solution $Q = UV^\top$ from Eq.~(6) without modifying it.
>
> To address this point, we also evaluated the method mentioned by the reviewer which flips the sign of the last column of $U$ whenever $\det(Q) < 0$ so that $\det(Q)=1$ is enforced. On Llama-7B, tested with both 128 and 512 calibration samples, this variant produced results that were almost identical to the unmodified solution: perplexity changed only at the level of the third decimal place, and no qualitative differences were observed. We believe this suggests that enforcing $\det(Q)=1$ has negligible practical effect under our settings.
>
> For clarity, we follow common practice and simply refer to $Q$ as a ``rotation'' throughout the paper, even though it may occasionally include a reflection.

---

### Author Response · Authors · 2025-11-21
**Short notice regarding the rebuttal**

First of all, we would like to express our sincere appreciation for your thoughtful reviews and for the committee’s effort in coordinating the discussion.
We are currently running the most essential additional experiments that directly address the key points raised in your comments.
To properly incorporate these results into the rebuttal, we expect it to be ready in about 3–4 days.

Thank you very much for your patience and understanding.
Authors

---

### Author Response · Authors · 2025-12-03
**Rebuttal Summary**

We conducted extensive additional experiments and analyses to address the major concerns raised by reviewers. These results demonstrate robustness across calibration sizes, strong performance relative to structured pruning baselines, scalability to larger models, and the practical benefit of the scaled variant.
For questions specific to individual reviewers, please refer to our responses to each reviewer.
The discussed issues have already been incorporated into the revised manuscript.

1. [SQAA - Weakness5/Question2, hBF1 - W2, hZjV - W1/Q1]

The reviewers pointed out that all experiments use only 128 samples. They requested both a comparison between RCPU and ridge regularization, and additional quantitative assessments of RCPU’s robustness across different calibration sizes.

In response, we added the experiments with different calibration sizes.
In those, we compared RCPU with the ridge-regularized least square method as well.

By [Figure2](https://anonymous.4open.science/r/anonymous-llm-pruning-D884/figure2.png), we confirmed that larger calibration sets stabilize performance. In addition, we showed the robustness of RCPU across calibration sizes in [Table2](https://anonymous.4open.science/r/anonymous-llm-pruning-D884/table2.png), [Table6](https://anonymous.4open.science/r/anonymous-llm-pruning-D884/table6.png), [Table7](https://anonymous.4open.science/r/anonymous-llm-pruning-D884/table7.png), and [Table3](https://anonymous.4open.science/r/anonymous-llm-pruning-D884/table3.png).
At the same time, we showed RCPU's superior downstream performance over the ridge-regularized least square method. We confirmed that the performance of ridge-regularized least square method is inferior to RCPU in terms of benchmark accuracy.
Moreover, we added the analysis for the hyper-parameter of the ridge regularization in [Figure5](https://anonymous.4open.science/r/anonymous-llm-pruning-D884/figure5.png) and [Figure6](https://anonymous.4open.science/r/anonymous-llm-pruning-D884/figure6.png).
The best hyper-parameter varies widely depending on the model type, pruning ratio, and the number of calibration samples. This suggests that finding an optimal one in a reliable manner is inherently difficult.

2. [SQAA – W1, hBF1 – W1, hZjV - W4]

The reviewers pointed out that our comparison to structure-preserving methods such as SliceGPT was limited in the experiments.

Conducting additional experiments, we compared RCPU with SliceGPT. [Table3](https://anonymous.4open.science/r/anonymous-llm-pruning-D884/table3.png) and [Table7](https://anonymous.4open.science/r/anonymous-llm-pruning-D884/table7.png) show that RCPU maintains better accuracy on Llama-2-13B, especially at high ratios. We explained that SliceGPT applies a global orthogonal transform followed by global pruning, which can lead to distribution mismatches across layers, reducing accuracy at high pruning ratio. In contrast, RCPU's layer-wise approach aligns each layer more closely with its input distribution, resulting in more stable performance.

3. [hBF1 – W1, hZjV – W3/W4, f6Ag – W2/Q2]

These comments all highlighted that the experiments are limited to the LLaMA-7B model, with a lack of evaluation on larger models such as 13B.

We demonstrated RCPU's effectiveness on a larger model (Llama-2-13B), as shown in [Table1](https://anonymous.4open.science/r/anonymous-llm-pruning-D884/table1.png), [Table3](https://anonymous.4open.science/r/anonymous-llm-pruning-D884/table3.png), and [Table 7](https://anonymous.4open.science/r/anonymous-llm-pruning-D884/table7.png). This confirms that RCPU scales effectively beyond the 7B model, maintaining its advantages in perplexity and downstream task accuracy across different model sizes.
We also conducted evaluations on Vicuna-7B and obtained similar results shown in [Table8](https://anonymous.4open.science/r/anonymous-llm-pruning-D884/table8.png), [Table9](https://anonymous.4open.science/r/anonymous-llm-pruning-D884/table9.png), and [Table10](https://anonymous.4open.science/r/anonymous-llm-pruning-D884/table10.png).

4. [SQAA – W4, hBF1 – Q3]

These comments raised concerns about the choice and effect of the scaled variant, with reviewers questioning the usefulness of employing a single global scale $s$.

We conducted additional experiments using 128 and 512 calibration samples to evaluate the effectiveness of the scaled variant. [Table2](https://anonymous.4open.science/r/anonymous-llm-pruning-D884/table2.png) and [Table6](https://anonymous.4open.science/r/anonymous-llm-pruning-D884/table6.png) ([Table7](https://anonymous.4open.science/r/anonymous-llm-pruning-D884/table7.png) and [Table3](https://anonymous.4open.science/r/anonymous-llm-pruning-D884/table3.png)) demonstrate that the scaled variant performs better with larger calibration sets. The authored explained that the additional samples stabilize the norm of the unpruned versus pruned outputs, allowing the global scale $s^\star$ to more effectively restore the original magnitude.

---

### Meta-Review · Area_Chair_AvDr · 2026-01-07

**Summary:**

The reviewers raised concerns primarily around the scope and robustness of the experimental evaluation, including reliance on a single calibration size (128 samples), limited evaluation on only LLaMA-7B in the initial submission, and lack of comparison with regularized least-squares approaches and other structure-preserving pruning methods such as SliceGPT. Some reviewers also questioned whether the claimed advantage of rotation constraints over standard regularization was sufficiently justified, and whether the observed gains were practically meaningful given the remaining gap to the dense model.
At the same time, reviewers consistently acknowledged the soundness and clarity of the proposed method, highlighting the clean orthogonal Procrustes formulation, the geometry-preserving motivation, and the practical appeal of a training-free, layer-wise compensation approach. The suggested decision is informed by the authors’ substantial rebuttal, which meaningfully strengthened the empirical evidence and addressed the most critical concerns regarding robustness, scalability, and baseline comparisons.

**Reviewer Concerns:**

Concerns addressed by the rebuttal:
1. Limited calibration size analysis: The authors added experiments across multiple calibration sizes (including 512 samples), demonstrating stability and clarifying when the scaled variant is beneficial.
2. Comparison with regularized least squares: Direct comparisons with ridge-regularized LS were added, along with an analysis of overfitting via degrees of freedom.
3. Model scale and generality: Additional evaluations on larger models (LLaMA-2-13B and Vicuna-7B) were provided, addressing concerns about scalability beyond LLaMA-7B.
4. Comparison with SliceGPT: New experimental results and discussion clarified differences between global and layer-wise orthogonal transformations.
5. Implementation details: Numerical aspects of the Procrustes solution (e.g., reflections vs. enforced rotations) and code availability were clarified.

Concerns partially or not fully addressed:
1. Quantitative measurement of “geometry preservation” (e.g., explicit cosine similarity or norm diagnostics) remains limited.
2. Broader comparisons with a wider range of pruning or compression methods could further contextualize the gains.
3. Some reviewers remain unconvinced about the overall practical impact due to the residual performance gap to dense models, although this is a known limitation of structured pruning in general.

**Reviewer Scores:**

Reviewer SQAA: Reviewer SQAA would likely maintain or slightly increase their score (e.g., from 6 to 6–7), as most requested analyses and comparisons were explicitly addressed in the rebuttal.
Reviewer hBF1: Reviewer hBF1 would likely increase their score modestly (e.g., from 6 to 7), given the added experiments on larger models, calibration sizes, and additional baselines.
Reviewer hZjV: Reviewer hZjV may increase slightly but remain cautious (e.g., from 2 to 3–4). While robustness, ridge regression comparisons, and larger-model evaluations were addressed, this reviewer expressed fundamental skepticism about the practical significance of the gains, which may not be fully resolved.
Reviewer f6Ag: Reviewer f6Ag maintained the positive score, as the rebuttal largely addressed and resolved the concerns he had raised.

---

### Decision · Program_Chairs · 2026-01-26

Accept (Poster)